# A six-year-long (2013–2018) high-resolution air quality reanalysis dataset in China based on the assimilation of surface observations from CNEMC

Lei Kong[1,2], Xiao Tang[1,2], Jiang Zhu[1,2], Zifa Wang[1,2,3], Jianjun Li[4], Huangjian Wu[1,5], Qizhong Wu[6], Huansheng Chen[1],   Lili Zhu[4], Wei Wang[4], Bing Liu[4], Qian Wang[7], Duohong Chen[8], Yuepeng Pan[1,2], Tao Song[1,2], Fei Li[1], Haitao Zheng[9], Guanglin Jia[10], Miaomiao Lu[11], Lin Wu[1,2], Gregory R. Carmichael[12]

[1]LAPC & ICCES, Institute of Atmospheric Physics, Chinese Academy of Sciences, Beijing, 100029, China
[2]College of Earth and Planetary Sciences, University of Chinese Academy of Sciences, Beijing, 100049, China
[3]Center for Excellence in Regional Atmospheric Environment, Institute of Urban Environment, Chinese Academy of Sciences, Xiamen 361021, China
[4]China National Environmental Monitoring Centre, Beijing, 100012, China
[5]Guanghua School of Management, Peking University, Beijing 100871, China
[6]College of Global Change and Earth System Science, Beijing Normal University, Beijing 100875, China
[7]Shanghai Environmental Monitoring Center, Shanghai, 200030, China
[8]State Environmental Protection Key Laboratory of Regional Air Quality Monitoring, Guangdong Environmental Monitoring Center, Guangzhou, 510308, China
[9]Key Lab of Environmental Optics and Technology, Anhui Institute of Optics and Fine Mechanics, Hefei Institutes of Physical Science, Chinese Academy of Sciences, Hefei, 230031, China
[10]School of Environment and Energy, South China University of Technology, University Town, Guangzhou 510006, China
[11]State Environmental Protection Key Laboratory of Urban Ambient Air Particulate Matter Pollution Prevention and Control, College of Environmental Science and Engineering, Nankai University, Tianjin 300350, China
[12]Center for Global and Regional Environmental Research, University of Iowa, Iowa City, IA, 52242, USA

*Correspondence to*: Xiao Tang (tangxiao@mail.iap.ac.cn) and Jiang Zhu (jzhu@mail.iap.ac.cn)

## Abstract

A six-year-long high-resolution Chinese air quality reanalysis (CAQRA) dataset is presented in this study obtained from the assimilation of surface observations from China National Environmental Monitoring Centre (CNEMC) using the ensemble Kalman filter (EnKF) and Nested Air Quality Prediction Modeling System (NAQPMS). This dataset contains surface fields of six conventional air pollutants in China (i.e., $PM_{2.5}$, $PM_{10}$, $SO_2$, $NO_2$, CO and $O_3$) for period 2013–2018 at high spatial (15 km×15 km) and temporal (1 hour) resolutions. This paper aims to document this dataset by providing detailed descriptions of the assimilation system and the first validation results for the above reanalysis dataset. The fivefold cross-validation (CV) method is adopted to demonstrate the quality of the reanalysis. The CV results show that the CAQRA yields an excellent performance in reproducing the magnitude and variability of surface air pollutants in China from 2013 to 2018 (CV $R^2$ = 0.52–0.81, CV root mean square error (RMSE) = 0.54 mg/m$^3$ for CO and CV RMSE = 16.4–39.3 μg/m$^3$ for the other pollutants at the hourly scale). Through comparison to the Copernicus Atmosphere Monitoring Service reanalysis (CAMSRA) dataset produced by the European Centre for Medium-Range Weather Forecasts (ECWMF), we show that CAQRA attains a high

accuracy in representing surface gaseous air pollutants in China due to the assimilation of surface observations. The fine horizontal resolution of CAQRA also makes it more suitable for air quality studies at the regional scale. The PM$_{2.5}$ reanalysis dataset is further validated against the independent datasets from the U.S. Department State Air Quality Monitoring Program over China, which exhibits a good agreement with the independent observations ($R^2$ = 0.74–0.86 and RMSE =16.8–33.6 μg/m$^3$ in different cities). Besides, through the comparison to satellite estimated PM$_{2.5}$ concentrations, we show that the accuracy of the PM$_{2.5}$ reanalysis is higher than that of most satellite estimates. The CAQRA is the first high-resolution air quality reanalysis dataset in China that simultaneously provides the surface concentrations of six conventional air pollutants, which is of great value for many studies, such as health impact assessment of air pollution, investigation of air quality changes in China, model evaluation and satellite calibration, optimization of monitoring sites and provision of training data for statistical or artificial intelligence (AI)-based forecasting. All datasets are freely available at https://doi.org/10.11922/sciencedb.00053 (Tang et al., 2020a), and a prototype product containing the monthly and annual means of the CAQRA dataset has also been released at https://doi.org/10.11922/sciencedb.00092 (Tang et al., 2020b) to facilitate the potential evaluation of the CAQRA dataset by users.

## 1 Introduction

Air pollution is a critical environmental issue that adversely affects human health and is closely connected to climate change (von Schneidemesser et al., 2015). Exposure to ambient air pollution has been confirmed by many epidemiological studies to be a leading contributor to the global disease burden, which increases both morbidity and mortality (Cohen et al., 2017). China, as the largest developing country, has achieved great economic development since the 1980s. This large-scale economic expansion, however, is accompanied by a dramatic increase in air pollutant emissions, leading to severe air pollution in China (Kan et al., 2012). Since 2012, the Chinese government has established a nationwide ground-based air quality monitoring network (Fig. 1) to monitor the surface concentrations of six conventional air pollutants in China, i.e., particles with an aerodynamic diameter of 2.5 μm or smaller (PM$_{2.5}$), particles with an aerodynamic diameter of 10 μm or smaller (PM$_{10}$), sulfur dioxide (SO$_2$), nitrogen dioxide (NO$_2$), carbon monoxide (CO) and Ozone (O$_3$), which plays an irreplaceable role in understanding the air pollution in China. In addition, since the implementation of Action Plan for the Prevention and Control of Air Pollution in 2013, a series of aggressive control measures has been applied in China to reduce the emissions of air pollutants. According to the estimates of Zheng et al. (2018b), the Chinese anthropogenic emissions has decreased by 59% for SO$_2$, 21% for NO$_x$, 23% for CO, 36% for PM$_{10}$ and 35% for PM$_{2.5}$ from 2013 to 2017. Concurrently, the air quality in China has changed dramatically over the past six years (Silver et al., 2018; Zheng et al., 2017). Such large changes in the Chinese air quality and their effects on human health and the environment have become an increasingly hot topic in many scientific fields (e.g., Xue et al., 2019; Zheng et al., 2017) which requires a long-term air quality dataset in China with high accuracy and spatiotemporal resolutions.

Ground-based observations can provide accurate information on the spatial and temporal distributions of air pollutants in
China, but they are sparsely and unevenly distributed in space. Satellite observations exhibit the advantages of a high spatial
coverage and have widely been applied in air pollution monitoring over large domains. A series of satellite retrievals related
to air quality has been developed over the past two decades, such as the observations of $NO_2$, $SO_2$ and $O_3$ columns from the
Ozone Monitoring Instrument (OMI; Levelt et al., 2006), CO column observations from the Measurement of Pollution in the
Troposphere (MOPITT; Deeter et al., 2003) and aerosol optical depth (AOD) observations from the Moderate Resolution
Imaging Spectroradiometer (MODIS; Barnes et al., 1998). The satellite column measurements have also been used to estimate
surface concentrations using different methods, such as chemical transport models (CTMs) (e.g., van Donkelaar et al., 2016;
van Donkelaar et al., 2010), advanced statistical methods (e.g., Ma et al., 2014; Ma et al., 2016; Xue et al., 2019; Zou et al.,
2017) and semi-empirical models (e.g., Lin et al., 2015; Lin et al., 2018), which have been proven to be an effective way to
acquire wide-coverage distributions of surface air pollutant with a good accuracy (Chu et al., 2016; Shin et al., 2019). However,
challenges remain in satellite-based concentration estimates due to missing values related to cloud contamination, uncertainties
in satellite measurements, and difficulties in modelling the complex relationship between surface concentrations and column
measurements (Shin et al., 2019; van Donkelaar et al., 2016; Xue et al., 2019). In addition, most satellite-based estimates of
surface concentrations exhibit low temporal resolutions (daily or even longer), which limits their application in fine-scale
studies, such as the assessment of the acute health effects of the air quality. To our knowledge, a nationwide long-term estimate
of the surface concentrations of all conventional air pollutants in China at the hourly scale have not yet been reported in
previous satellite estimates.
A long-term air quality reanalysis dataset of critical air pollutants can provide constrained estimates of their concentrations
at all locations and times, which optimally combines the accuracy of observations and the physical information and spatial
continuity of CTMs through advanced data assimilation techniques. Reanalysis datasets are uniform, continuous and state-of-
science best-estimate data products that have been adopted by a vast number of research communities. For example, several
long-term meteorological reanalysis datasets have been developed by various weather centres in different regions/countries,
such as the ERA-Interim reanalysis developed by the European Centre for Medium-Range Weather Forecasts (ECMWF; Dee
et al., 2011), the National Center for Atmospheric Research (NCAR)/National Centers for Environmental Protection (NCEP)
reanalysis developed by the NCEP (Saha et al., 2010), the Modern-Era Retrospective Analysis for Research and Applications
(MERRA) developed by the NASA Global Modeling and Assimilation Office (NASA-GMAO; Rienecker et al., 2011), the
Japanese 55-year Reanalysis (JRA-55) developed by the Japan Meteorological Agency (Kobayashi et al., 2015) and the China
Meteorological Administration's global atmospheric Reanalysis (CRA-40) developed by the China Meteorological
Administration (CMA). The use of data assimilation in atmospheric chemistry reanalysis is more recent, and certain reanalysis
datasets for the atmospheric composition have been produced over the past decades, for example the Monitoring Atmospheric
Composition and Climate (MACC), Copernicus Atmosphere Monitoring Service (CAMS) interim reanalysis (CIRA), and
CAMS reanalysis (CAMSRA) produced by the ECWMF (Flemming et al., 2017; Inness et al., 2019; Inness et al., 2013), the
MERRA-2 aerosol reanalysis produced by the NASA-GMAO (Randles et al., 2017), the tropospheric chemistry reanalysis
(TCR) from 2005–2012 produced by Miyazaki et al. (2015) and its latest version TCR-2 (Miyazaki et al., 2020), the global
reanalysis of carbon monoxide produced by Gaubert et al. (2016), the multi-sensor total ozone reanalysis from 1970–2012
produced by van der A et al. (2015) and the Japanese Reanalysis for Aerosols (JRAero) from 2011–2015 produced by
Yumimoto et al. (2017). These reanalysis datasets promote our understanding of the atmospheric composition and also
facilitate the air quality research. However, these datasets are all global datasets with coarse horizonal resolutions (> 50 km),
which may be insufficient to capture the high spatial variability of air pollutants at the regional scale. In addition, some of
these reanalysis datasets only provide air quality data prior to 2012 and only focus on specific species. There is still no high-
resolution air quality reanalysis dataset in China capturing its dramatic air quality change during recent years.

In view of these discrepancies, in this study, we develop a high-resolution regional air quality reanalysis dataset in China

from 2013 to 2018 (which will be extended in the future on a yearly basis) by assimilating surface observations from China
National Environmental Monitoring Centre (CNEMC). The developed reanalysis dataset may help mitigate the lack of high-
resolution air quality datasets in China by providing surface concentration fields of all six conventional air pollutants in China
at high spatial (15 km×15 km) and temporal (hourly) resolutions, which is of great value to (1) retrospective air quality analysis
in China, (2) health and environmental impact assessment of air pollution at fine scales, (3) model evaluation and satellite
calibration, (4) optimization of monitoring sites and (5) provision of basic training datasets for statistical or artificial
intelligence (AI)-based forecasting.
**2 Description of the chemical data assimilation system**

The Chinese air quality reanalysis (CAQRA) dataset was produced with the chemical data assimilation system

(ChemDAS) developed by the Institute of Atmospheric Physics, Chinese Academy of Sciences (IAP, CAS) (Tang et al., 2011).
This system consists of (i) a three-dimensional CTM called the Nested Air Quality Prediction Modeling System (NAQPMS)
developed by Wang et al. (2000), (ii) an ensemble Kalman filter (EnKF) assimilation algorithm, and (iii) surface observations
from CNEMC with the automatic outlier detection method. We adopted an offline analysis scheme in this study since there
are no previous experiences with online chemical data assimilation at such a high horizontal resolution. The lessons learned
from this offline analysis application could also facilitate future implementation of online analysis. In the offline analysis
scheme, a free ensemble simulation was first conducted, and the observations were then assimilated using the EnKF. A similar
offline analysis scheme has also been applied in previous reanalysis studies, such as Candiani et al. (2013) and Kumar et al.
(2012). Detailed descriptions of the ensemble simulation, observations and data assimilation algorithm used in this study are
presented below.
**2.1 Air pollution prediction model**

The NAQPMS model was used as the forecast model to represent the atmospheric chemistry, which has been applied in

previous assimilation studies (Tang et al., 2011; Tang et al., 2013). The model is driven by the hourly meteorological fields
produced by the Weather Research and Forecasting (WRF) model (Skamarock, 2008). Gas phase chemistry is simulated with
the carbon bond mechanism Z (CBM-Z) developed by Zaveri and Peters (1999). Aqueous-phase chemistry and wet deposition
are simulated based on the Regional Acid Deposition Model (RADM) mechanism in the Community Multi-scale Air Quality
(CMAQ) model version 4.6. In regard to aerosol processes, the thermodynamic model ISORROPIA 1.7 (Nenes et al., 1998)
is applied for the simulations of inorganic atmospheric aerosols. Six secondary organic aerosols (SOAs) are explicitly treated
in the NAQPMS model based on Li et al. (2011). To simulate the interactions between particles and gases, 28 heterogeneous
reactions involving sulfate, soot, dust and sea salt particles are included based on previous studies (Li et al., 2015; Li et al.,
2012). Size-resolved mineral dust emissions are calculated online as a function of the relative humidity, frictional velocity,
mineral particle size distribution and surface roughness (Li et al., 2012). Sea salt emissions are calculated with the scheme of
Athanasopoulou et al. (2008). The dry deposition of gases and aerosols is modelled based on the scheme of Wesely (1989),
and advection is simulated with the accurate mass conservation algorithm of Walcek and Aleksic (1998).
Figure 1 shows the modelling domain of this study, which covers most parts of East Asia with a fine horizontal resolution
of 15 km. The vertical coordinate system consists of 20 terrain-following levels with the model top reaching up to 20000 m
and the first layer at approximately 50 m. Nine vertical layers are set within 2 km of the surface to better characterize the
vertical mixing process within the boundary layers. The emissions of air pollutants considered in this study include the monthly
anthropogenic emissions retrieved from the Hemispheric Transport of Air Pollution (HTAP) v2.2 emission inventory with a
base year of 2010 (Janssens-Maenhout et al., 2015), biomass burning emissions retrieved from the Global Fire Emissions
Database (GFED) version 4 (Randerson et al., 2017; van der Werf et al., 2010), biogenic volatile organic compound (BVOC)
emissions retrieved from the Model of Emissions of Gases and Aerosols from Nature (MEGAN)-MACC (Sindelarova et al.,
2014), marine VOC emissions retrieved from the POET database (Granier et al., 2005), soil $NO_x$ emissions retrieved from the
Regional Emission Inventory in Asia (Yan et al., 2003) and lightning NOx emissions retrieved from Price et al., 1997. Clean
initial conditions are used in the air quality simulations with a two-week free run of the NAQPMS model as the spin-up time.
The top and boundary conditions are provided by the Model for Ozone and Related Chemical Tracers (MOZART; Brasseur et
al., 1998; Hauglustaine et al., 1998) model, and the meteorological fields are provided by the WRF model. In each daily
meteorology simulation, a 36-h free run of the WRF model is conducted with the first 12-h simulation period as the spin-up
run and the remaining 24-h period providing the meteorologic inputs for the NAQPMS model. The initial and boundary
conditions for the meteorology simulations are provided by the NCAR/NCEP 1° × 1° reanalysis data.
**2.2 Generation of ensemble simulation**
The EnKF uses an ensemble of model simulation to represent the forecast uncertainty which should include the most
model uncertain aspects. Considering that the emissions are a major source of uncertainty in air quality prediction (Carmichael
et al., 2008; Hanna et al., 1998; Li et al., 2017), in this study the ensemble were generated by perturbing the emissions based
on their error probability distribution functions (PDFs) which were assumed to be Gaussian distributions. Table 1 lists the
perturbed species considered in this study as well as their corresponding emission uncertainties obtained from previous studies.
The perturbed emissions were parameterized by multiplying the base emissions with a perturbation factor $\boldsymbol{\beta}$, as expressed in
Eq. (1):
$\boldsymbol{E_i} = \boldsymbol{E} \circ \boldsymbol{\beta_i}, i = 1, 2, \cdots, N$           (1)
where $\boldsymbol{E}$ denotes the vector of base emissions, $\circ$ denotes the Schur product and $N$ denotes the ensemble size. The performance
of the EnKF is strongly related to the ensemble size which determines the accuracy to which the background error covariance
is approximated (Constantinescu et al., 2007; Miyazaki et al., 2012). A large ensemble size is important in capturing the proper
background error covariance structure, especially in high-resolution data assimilation application due to the fine-scale
variability and large degree of freedoms. However, a large ensemble is computationally expensive as the cost of EnKF linearly
increases with ensemble size while the accuracy of covariance estimate improves by its square root (Constantinescu et al.,
2007). Thus, an appropriate ensemble should keep a good balance between accuracy and computational cost. Constantinescu
et al. (2007) in their ideal experiments showed that a 50-member ensemble has significant improvement against smaller
ensembles, and Miyazaki et al. (2012) in their real chemical assimilation experiments showed that the improvement was much
less significantly by further increasing the ensemble size from 48 to 64. Thus, the ensemble size was chosen as 50 in this study
by referencing pervious publications and also our previous high-resolution regional assimilation work (Tang et al., 2011; Tang
et al., 2013; Tang et al., 2016) which showed that a 50-member ensemble keeps good balance between assimilation
performance and computational efficiency. However, it should be noted that our application has higher horizontal resolution
than that of Constantinescu et al. (2007) and Miyazaki et al. (2012), which may require larger ensemble size due to the larger
degree of freedoms in our application. Thus, to reduce the degree of freedoms in our high-resolution data assimilation work,
we assumed that the emission errors were spatially correlated, and an isotropic correlation model was assumed in the
covariance of the emission errors, which is written as:
$\rho(i,j) = exp\left\{-\frac{1}{2}\left[\frac{h(i,j)}{l}\right]^2\right\}$           (2)
where $\rho(i,j)$ represents the correlation between grids $i$ and $j$, $h(i,j)$ is the distance between these two points and $l$ is the
decorrelation length, which was specified as 150 km in this study. According to the PDF of the emission errors, $\boldsymbol{\beta}$ follows the
same Gaussian distribution as that of the emission errors except that its mean equals 1. Using the method of Evensen (1994),
fifty smooth pseudorandom perturbation fields of $\boldsymbol{\beta}$ were generated for each perturbed species. In addition, the emission
perturbations were kept independent from each other to prevent pseudo-correlation among the different species.
**2.3 Observations**
Surface observations of the hourly ambient $PM_{2.5}$, $PM_{10}$, $SO_2$, $NO_2$, CO and $O_3$ concentrations retrieved from the CNEMC
were used in this study. The number of observation sites was approximately 510 in 2013 and increased to 1436 in 2015. Real-
time observations of these six air pollutants at each monitoring site are routinely gathered by the CNEMC and released to the
public (available at http://www.cnemc.cn/; last accessed: 17 April 2020) at hourly intervals. A challenge that should be
overcome in the assimilations of surface observation is that there are occasional outliers occurring in these observations due
to the instrument malfunctions, influences of harsh environments and limitation of measurement method. Filtering out these
outliers is necessary before the assimilation, otherwise these outliers may cause unrealistic spatial and temporal variations in
the reanalysis. To address this issue, a fully automatic outlier detection method was developed by Wu et al. (2018) to filter out
the observation outliers. An automatic outlier detection method is very important in chemical data assimilation since there is
a large amount of observation data on multiple species. Four types of outliers characterized by temporal and spatial
inconsistencies, instrument-induced low variances, periodic calibration exceptions and lower $PM_{10}$ concentrations than those
of $PM_{2.5}$ were detected and removed before the assimilation. Figure S1 shows the removal ratios of the six air pollutants from
2013 to 2018, which are generally around 1.5% for most air pollutants throughout the assimilation period. The $PM_{10}$
observations have a high removal ratio (9–13%) during 2013–2015 with most of these outliers marked by lower $PM_{10}$
concentrations than those of $PM_{2.5}$. However, there was a sharp decrease in removal ratios of $PM_{10}$ in 2016 (~1.5%) because
of the implementation of a compensation algorithm for the loss of semi-volatile materials in the $PM_{10}$ measurements (Wu et
al., 2018). To assess the potential impacts of outlier detection on the assimilations, the differences in annual concentrations
caused by quality control are shown in Fig. S2. The differences were generally positive for $PM_{2.5}$, $SO_2$, $NO_2$ and CO
concentrations, indicating a lower tendency of these species' concentrations due to the use of outlier detection. Negative
differences were mainly found in the $PM_{10}$ concentrations in south China and the $O_3$ concentrations throughout China.
According to estimation, the impacts of outlier detection were generally small in most stations. The differences were less than
5 μg/m$^3$ (1 μg/m$^3$) for $PM_{2.5}$ concentrations over most stations in north (south) China and less than 1 μg/m$^3$ for the gaseous
air pollutants for most stations throughout China. The differences were shown to be relative larger for $PM_{10}$ concentrations
over northwest China which can be over 20 μg/m$^3$ in stations around Taklimakan Desert. This would be due to the higher
outlier ratios in the observations over the remote areas. More details on the outlier detection method were available in Wu et
al. (2018).
A proper estimate of the observation error is important in regard to the filter performance since the observation and
background errors determine the relative weights of the observation and background values in the analysis. The observation
error includes measurement and representativeness errors. For each species, the measurement error was given by their
respective instruments, namely, 5% for $PM_{2.5}$ and $PM_{10}$, 2% for $SO_2$, $NO_2$ and CO, and 4% for $O_3$ according to officially
released documents of the Chinese Ministry of Ecology and Environmental Protection (HJ 193–2013 and HJ 654–2013,
available at http://www.cnemc.cn/jcgf/dqhj/; last accessed: 17 April 2020). The representativeness error arises from the
different spatial scales that the gridded model results and discrete observations represent, which is parameterized by the
formula proposed by Elbern et al., (2007) in this study:
$r_{repr} = \sqrt{\frac{\Delta x}{L_{repr}}} \times \epsilon^{abs}$ (3)
where $r_{repr}$ represents the representativeness error, $\Delta x$ represents the model resolution, $L_{repr}$ represents the characteristic
representativeness length of the observation site and $\varepsilon^{abs}$ represents the error characteristic parameters for different species.
The estimation of $L_{repr}$ is dependent on the types of observation sites with urban sites usually having smaller representative
length than the rural sites have due to the larger representativeness errors. Considering that the observation sites from CNEMC
were almost city (urban) sites (>90%), the $L_{repr}$ was assigned to be 2km in this study according to Elbern et al., 2007.

For the estimations of $\varepsilon^{abs}$, previous studies (Chen et al., 2019; Feng et al., 2018; Jiang et al., 2013; Ma et al., 2019;

Pagowski and Grell, 2012; Peng et al., 2017; Werner et al., 2019) usually assigned the $\varepsilon^{abs}$ empirically to be half of the
measurement error following the study by Pagowski et al. (2010). In this study, the $\varepsilon^{abs}$ was obtained from Li et al. (2019)
who estimated the $\varepsilon^{abs}$ based on a dense observation network in Beijing-Tianjin-Hebei region. In their study, the
representativeness error of each species' observation was first estimated by the spatiotemporal averaged standard deviation of
the observed values within a 30km×30km grid:
$r_{repr,i} = \frac{1}{MT}\sum_{m=1}^{M}\sum_{t=1}^{T}S_{m,t,i}$         (4)
where $r_{repr,i}$ represents the representativeness errors of the observations for species $i$, $S_{m,t,i}$ represents the standard deviation
of the observed values of species $i$ at different sites that are located in a same grid $m$ at time $t$, $M$ and $T$ represents the total
number of grid and observation time. After the estimations of $r_{repr,i}$, the $\varepsilon_i^{abs}$ for species $i$ were estimated by a transformation
of Eq. (3):
$\varepsilon_i^{abs} = r_{repr,i}/\sqrt{\frac{\Delta x}{L_{repr}}}$         (5)
where Δx is equal to 30km. Based on the estimated $L_{repr}$ and the $\varepsilon_i^{abs}$ for different species, the representativeness errors are
estimated using Eq. (3) by specifying the $\Delta x$ to be 15km.
**2.4 Data assimilation algorithm**

We used a variant of the EnKF approach, i.e., the local ensemble transform Kalman filter (LETKF; Hunt et al., 2007), to

assimilate the observations into the model state. The LETKF has several advantageous over the original EnKF (e.g., Miyazaki
et al., 2012). As a kind of deterministic filter, it does not need to perturb the observations, which avoids introducing additional
sampling errors. In addition, the LETKF performs the analysis locally in space and time, which not only alleviate the rank
problem of the EnKF method but also suppress the long-distance spurious correlation caused by the limited ensemble size.
The formulation of the LETKF can be written as:
$\overline{x^a} = \overline{x^b} + \mathbf{X^b}\overline{w}^a$         (6)
$\overline{w}^a = \widetilde{\mathbf{P}}^a\left(\mathbf{HX^b}\right)^T\mathbf{R}^{-1}(y^o - \mathbf{H}\overline{x^b})$         (7)
$\widetilde{\mathbf{P}}^a = \left[\frac{(N_{ens}-1)\mathbf{I}}{1+\lambda} + \left(\mathbf{HX^b}\right)^T\mathbf{R}^{-1}\left(\mathbf{HX^b}\right)\right]^{-1}$         (8)
$\overline{x^b} = \frac{1}{N_{ens}}\sum_{i=1}^{N_{ens}}x_i^b\ ;\mathbf{X_i^b} = \frac{1}{\sqrt{N-1}}\left(x_i^b - \overline{x^b}\right)$         (9)

where $\overline{x^a}$ is the analysis state, $\overline{x^b}$ is the background state, $\mathbf{X^b}$ represents the background perturbations, $\overline{w^a}$ is the analysis in the ensemble space spanned by $\mathbf{X^b}$, $\widetilde{\mathbf{P}}^{\mathbf{a}}$ is the analysis error covariance in the ensemble space with dimensions of $N_{ens} \times N_{ens}$, $\mathbf{y^o}$ is the vector of observations used in the analysis of this grid, $\mathbf{R}$ is the observation error covariance matrix, and $\mathbf{H}$ is the linear observational operator that maps the model space to the observation space. The scalar $\lambda$ in Eq. (8) denotes the inflation factor for the background covariance matrix, which was estimated with the algorithm proposed by Wang and Bishop (2003):

$$\lambda = \frac{(\mathbf{R}^{-1/2}d)^{\mathbf{T}}\mathbf{R}^{-1/2}d - p}{trace\{\mathbf{R}^{-1/2}\mathbf{H}P^b(\mathbf{R}^{-1/2}\mathbf{H})^{\mathbf{T}}\}} \tag{10}$$

$$\boldsymbol{d} = \boldsymbol{y^o} - \mathbf{H}\overline{x^b} \tag{11}$$

$$\mathbf{P}^b = \mathbf{X^b}(\mathbf{X^b})^{\mathbf{T}} \tag{12}$$

where $\boldsymbol{d}$ represents the residuals, $p$ is the number of observations, $\mathbf{P}^b$ is the ensemble-estimated background error covariance matrix, and the trace of the covariance matrix is used to approximate covariance on a globally averaged basis. The inflation is necessary for the ensemble-based assimilation algorithm since the ensemble-estimated background error covariance is very likely to underestimate the true background error covariance due to the limited ensemble size and occurrence of the model error (Liang et al., 2012). Without any treatment to prevent background error covariance underestimation, the model forecast would be overconfident and eventually result in filter divergence. Using Eq. (10), the hourly inflation factor was calculated for each species. In addition, the inflation factor was calculated locally in this study. Thus, the inflation factor used in this assimilation is not only species specific, but also varies with time and space, which reflects different error characteristics of the different species in different time and places.

Besides, the inter-species correlation was neglected in the background error covariance similar to previous chemical data assimilation studies (e.g., Inness et al., 2015; Inness et al., 2019; Ma et al., 2019) although Miyazaki et al, (2012) has shown the benefits of including correlations between the background errors of different chemical species. This is, on the one hand, to avoid the effects of the spurious correlation between non or weakly related variables. On the other hand, different from Miyazaki et al., 2012, this study concentrated on the assimilations of primary air pollutants (except of $O_3$) whose errors are more related to the errors in their emissions. Since the emission errors of these species were considered to be independent in this study (Sect. 2.2), thus the correlation between background errors of different species were generally near zero for most cases as shown in Figs. S3-4. The high correlations only occur in background errors of $PM_{2.5}$ and $PM_{10}$ as well as the $NO_2$ and $O_3$. The high positive correlation between $PM_{2.5}$ and $PM_{10}$ is just because $PM_{2.5}$ is a part of $PM_{10}$, and there would be redundant information in the observations of $PM_{2.5}$ and $PM_{10}$ concentrations, thus we did not include the correlation between the $PM_{2.5}$ and $PM_{10}$ concentrations in the assimilation. The negative correlation between the $O_3$ and $NO_2$ is due to the $NO_x$-OH-$O_3$ chemical reactions in the $NO_x$ saturated conditions that the increases of $NO_2$ concentrations would reduce the $O_3$ concentrations due to the enhanced NO titration effect. However, the relationship between $O_3$ and $NO_2$ concentrations is actually nonlinear depending on the $NO_x$ limited or saturated conditions (Sillman, 1999), and previous study by Tang et al. 2016 has shown the limitations of the EnKF under strong nonlinear relationships. The cross-variable data assimilations of $O_3$ and $NO_2$ may come

up with inefficient or even wrong adjustments. Considering the nonlinear relationship between the $O_3$ and $NO_2$ concentrations
and their unexpected effects on EnKF, we took a conservative way in the assimilations of $NO_2$ and $O_3$ by neglecting their error
correlations. This would also make different species be assimilated in a consistent way. Therefore, in this study each air
pollutant is assimilated independently by only using the observations of this pollutant.

Figure 2 shows the local scheme we used in the assimilation, where the plus and dot symbols indicate the centres of the

model grids and locations of the observation sites, respectively. In each model grid, only the observation sites located within
a $(2l + 1)$ by $(2l + 1)$ rectangular area centred at this model grid were considered in the calculations of its analysis. The cut-
off radius $l$ was chosen as 12 model grids, approximately 180 km at the 15-km horizontal resolution. The use of a cut-off
radius, however, could cause analysis discontinuities when an observation enters or leaves the local domain when moving
from one model grid to another (Sakov and Bertino, 2011). To increase the smoothness of the analysis state, following Hunt
et al. (2007), we artificially reduced the impact of the observations close to the boundary of the local domain by multiplying
the entries in $\mathbf{R^{-1}}$ by a factor decaying from one to zero with increasing distance of the observation from the central model
grid. The decay factors used in this study are calculated by:
$\rho(i) = exp\left\{-\frac{h(i)^2}{2L^2}\right\}$         (13)
where $\rho(i)$ is the decay factor for observation $i$, $h(i)$ is the distance between observation $i$ and the central model grid point,
and $L$ is the decorrelation length chosen as 80 km, smaller than the cut-off radius, to increase the smoothness of the analysis
state. Typically, only the state of the central model grid is updated and used to construct the global analysis field. However,
experience has shown that an observable discontinuity remains in the analysis over certain regions. To address this issue,
following the method of Ott et al. (2004), we simultaneously updated the state of a small patch ($l =1$) around the central model
grid (the updated region in Fig. 2) at each local analysis step. The final analysis of a given model grid was then obtained as the
weighted mean of all the analysis values of this model grid. A weighted mean was necessary since the analysis of the different
patches adopted different decay factors for the observation error. The weight of each analysis value in model grid $i$ is calculated
by Eq. (14):
$W_{i,j} = \dfrac{\exp\left(-\frac{h(i,j)^2}{L^2}\right)}{\sum_{j=1}^{m} \exp\left(-\frac{h(i,j)^2}{L^2}\right)}$         (14)
where $h(i,j)$ is the distance of model grid $i$ to the central model grid of the patch generating the $j$th analysis value of this grid,
$m$ is the number of patches containing this model grid and $L$ is the decorrelation length, which was chosen as 80 km in this
study.

## 3 Data assimilation statistics

### 3.1 $\chi^2$ diagnosis

We first applied the $\chi^2$ test to demonstrate the performance of our data assimilation system, which is important in evaluating the reanalysis (Miyazaki et al., 2015). The $\chi^2$ diagnosis is a robust criterion for validating the estimated background and observation error covariance in the data assimilation (e.g. Menard et al., 2000; Miyazaki et al., 2015; Miyazaki et al., 2012), which is estimated by comparing the sample covariance of observation minus forecast (OmF) with the sum of estimated background and observation error covariance in the observational space ($\mathbf{HBH^T + R}$):

$$Y = \frac{1}{\sqrt{m}}(\mathbf{HBH^T + R})^{-\frac{1}{2}}(y^o - HX^b) \qquad (15)$$

$$\chi^2 = Y^T Y \qquad (16)$$

where $m$ is the number of observations. According to the Kalman filtering theory, the mean of $\chi^2$ should approach 1 if the background and observation error covariances are properly specified, while values greater (lower) than 1 indicates the underestimation (overestimation) of the observation and/or background error covariance.

Figure 3 shows the time series of the monthly $\chi^2$ values (black lines) for different species as well as the number of assimilated observations per month (blue bars). The mean values of $\chi^2$ are generally within 50% difference from the ideal value of 1 for $PM_{2.5}$, $PM_{10}$, $NO_2$ and $O_3$, which suggests that the observation and background error covariance are generally well specified in the analysis of these species. Although the $\chi^2$ values for these species showed pronounced seasonal variations that reflect the different error characteristics in different seasons, the $\chi^2$ values were roughly stable for $PM_{2.5}$ and $O_3$ throughout the assimilation periods, and for $NO_2$ and $PM_{10}$ after 2015 when the number of assimilated observations become stable, which generally shows the long-term stability of the performance of data assimilation. The $\chi^2$ values for $SO_2$ were nevertheless greater than 1 in most cases, especially before 2017. This would be more relevant to the underestimations of background error covariance of $SO_2$ as we only specified 12% uncertainty in the $SO_2$ emissions, suggesting that the emission uncertainty of $SO_2$ may be underestimated by Zhang et al. (2009). There were also pronounced annual trends in the $\chi^2$ values of $SO_2$, which may be attributed to the increases of observation number from 2013 to 2014 and the substantial decreases of $SO_2$ observations from 2013 to 2018. Although smaller than the $\chi^2$ values of $SO_2$, the values for CO were greater than 1 in most cases, suggesting the underestimations of the error covariances. Similar to the $\chi^2$ values of $SO_2$, obvious decreasing trend can also be found in the $\chi^2$ values of CO. These results suggest that our data assimilation system has relatively poor performance in the analysis of CO and $SO_2$ concentrations than the other four species, which is consistent with the cross-validation results (Sect. 4.2.2) that showed smaller $R^2$ values for the reanalysis data of CO and $SO_2$ concentrations. The annual trend of $\chi^2$ values in CO and $SO_2$ also indicates relatively weak stability in the performance of data assimilation system on assimilating CO and $SO_2$ observations, which may influence the analysis of the annual trends in these two species.

## 3.2 OmF & OmA analysis

Spatial distributions of six-year averaged OmF and observation minus analysis (OmA) for each species in the observation space were then analysed to investigate the structure of forecast bias and to measure the improvement in the reanalysis (Fig. 4). The analysis increment, which is estimated from the differences between the analysis and forecast, is also plotted to measure the adjustments made in the model space. The OmF values have showed persistent positive model biases (i.e., negative OmF) in the $PM_{2.5}$ and $SO_2$ concentrations in east China, as well as $PM_{10}$ and $O_3$ concentrations in south China. The negative model biases (i.e., positive OmF) were mainly found in the $PM_{2.5}$ concentrations in west China, the $PM_{10}$ concentrations in north China, the $O_3$ concentrations in central-east China, as well as the concentrations of CO and $NO_2$ throughout the whole China.

The OmA values suggest that the data assimilation removes most of the model biases for each species, which confirms the good performance of our data assimilation system. According to Fig. S5, the monthly mean OmF biases were almost completely removed in each regions of China because of the assimilation, with mean OmF biases reducing by 32–94% for $PM_{2.5}$, 33–83% for $PM_{10}$, 25–96% for $SO_2$, 53–88% for $NO_2$, 88–97% for CO and 54–90% for $O_3$ concentrations in different regions of China. The mean OmF root mean square error (RMSE) were also reduced substantially by 80–93% for $PM_{2.5}$, 80–86% for $PM_{10}$, 73–96% for $SO_2$, 76–91% for $NO_2$, 88–96% for CO and 76–87% for $O_3$ concentrations in different regions of China (Fig. S6). In addition, despite the mean OmF bias and OmF RMSE exhibit significant annual trend, the OmA bias and OmA RMSE are relatively stable during the assimilation period, which generally confirms the long-term stability of our data assimilation system.

The spatial patterns of analysis increment were in good agreement with those of the OmF values for each species, which generally shows negative (positive) increments for $PM_{2.5}$ concentrations in east (west) China, negative (positive) increments for $PM_{10}$ concentrations in south (north) China, negative increments for $SO_2$ throughout the China, positive increments for CO and $NO_2$ concentrations throughout the China, and the positive (negative) increments for $O_3$ concentrations in central-east (south) China. These results confirm that the data assimilation can effectively propagate the observation information into the model state and reduce the model errors.

## 4 Evaluation Results

In this section, we present the fields of the CAQRA dataset and compare them to the observations. It aims to provide a brief introduction to the CAQRA dataset and gives a first assessment of the quality of this dataset. The cross-validation (CV) method was applied in the assessment of the CAQRA dataset, in which a proportion of the observation data was withheld from the data assimilation process and adopted as a validation dataset. We conducted five CV experiments by randomly dividing the observation sites of the CNEMC into five groups (with 20% of the observation sites in each group). In each experiment, the analysis was performed with one group of the observation data omitted in the assimilation process. Analysis results at the validation sites, i.e., the observation sites not used in the assimilation process, were then collected and used to validate the assimilation results. For convenience, the analysis results at the validation sites of the five CV experiments were combined

and comprised a validation dataset containing all observation sites (the CV run). This dataset was then evaluated against the
observations to assess the quality of the CAQRA dataset. In addition, independent $PM_{2.5}$ observations retrieved from the U.S.
Department State Air Quality Monitoring Program over China were also employed in the assessment of the $PM_{2.5}$ reanalysis
field. The quality of the CAQRA dataset was assessed at different spatial and temporal scales to better understand the CAQRA
dataset. Additionally, the validation results of the ensemble mean of the simulations without assimilation (the base simulation)
are provided to highlight the impacts of assimilation.

## 4.1 Particulate matter (PM)

### 4.1.1 Spatial distribution of the PM reanalysis data over China

We first present the reanalysis fields of the PM concentrations ($PM_{2.5}$ and $PM_{10}$) in China. Figure 5 shows the six-year
mean (2013–2018) spatial distribution of the $PM_{2.5}$ concentration in China obtained from the CAQRA dataset, base simulation
and observations. The CAQRA dataset provides a continuous map of the $PM_{2.5}$ concentration in China and suitably reproduces
the observed magnitude of the $PM_{2.5}$ concentration in China. The highest $PM_{2.5}$ concentrations were observed in the NCP
region due to its intensive industrial activities and the associated high emissions of $PM_{2.5}$ and its precursors (Qi et al., 2017).
High $PM_{2.5}$ concentrations were also found in the SE region, where the $PM_{2.5}$ concentration is influenced by both local
emissions and the long-range transport of air pollutants from northern China (Lu et al., 2017). In the NW region, in addition
to hotspots exhibiting high $PM_{2.5}$ concentrations in large cities, high $PM_{2.5}$ concentrations were also observed in the Taklimakan
Desert due to the influences of dust emissions. The observed magnitude and spatial variability of the $PM_{10}$ concentration were
also represented well by the $PM_{10}$ reanalysis field. In general, the spatial distributions of the $PM_{10}$ reanalysis were similar to
those of the $PM_{2.5}$ reanalysis except in Gansu and Ningxia provinces, where high $PM_{10}$ concentrations and relatively low $PM_{2.5}$
concentrations occurred. This may be related to the large contributions of dust emissions in these areas. The base simulation
notably overestimated the $PM_{2.5}$ and $PM_{10}$ concentrations in China. This may occur due to the systematic biases in the emission
inventory (Kong et al., 2019) and because negative trends of PM and its precursor emissions were not considered in our
simulations. In addition, the $PM_{2.5}$ concentration hotspots in the NW region and Tibetan Plateau were not captured in the base
simulation, possibly due to the absence of emissions in these remote regions.
Seasonal maps of the $PM_{2.5}$ and $PM_{10}$ concentrations are shown in Figs. S7–8 in the Supplement, which reveal profound
seasonal variations. Both the $PM_{2.5}$ and $PM_{10}$ concentrations exhibit maximum values in winter in most regions of China due
to the increased anthropogenic emissions related to enhanced power generation, industrial activities and fossil fuel burning for
heating purposes (Li et al., 2017). Unfavourable meteorological conditions with stable boundary conditions also contribute to
the high PM concentrations in winter. In contrast, due to the low emission rate and intense mixing processes, the PM
concentrations are the lowest in summer. The PM concentrations in the Taklimakan Desert exhibit a different seasonality, with
the highest PM concentrations occurring in spring and the lowest levels occurring in winter. This occurs because the major
PM sources in the Taklimakan Desert are not anthropogenic emissions but dust emissions, which are usually the highest in

spring due to the frequent strong dust storms. Figure 6 further shows an example of the hourly PM reanalysis results, including a year-round time series of the site mean hourly PM concentrations in Beijing. This figure shows that PM reanalysis suitably captures the hourly evolution of the PM concentrations. Both the heavy haze episodes during the wintertime and the strong dust storms during the springtime are represented well in PM reanalysis.

### 4.1.2 Assessment of the PM reanalysis data over China

The CV method was used to assess the quality of the PM reanalysis data over China. Table 2 summarizes the site-based CV results for the reanalysis data from 2013 to 2018 at the different temporal scales. It should be mentioned that these sites are all validation sites not used in the data assimilation process. The validation results indicated that by assimilating the surface PM concentrations, the reanalysis data exhibit a relatively high performance in reproducing the magnitude and variability of the surface PM concentrations in China. The CV $R^2$ values were up to 0.81 and 0.72 in regard to the hourly $PM_{2.5}$ and $PM_{10}$ concentrations, respectively, which were much higher than the values of 0.26 and 0.17, respectively, in the base simulation. The bias was substantially reduced in the $PM_{2.5}$ and $PM_{10}$ reanalysis data with CV mean bias (MBE) values of approximately -2.6 μg/m$^3$ (-4.9%) and -6.8 μg/m$^3$ (-8.7%), respectively, at the hourly scale, much smaller than the large bias in the base simulation. The CV RMSE values were only approximately half of the base simulation RMSE values, which were approximately 17.6 and 39.3 μg/m$^3$ for the hourly $PM_{2.5}$ and $PM_{10}$ concentrations, respectively. The reanalysis data showed a good performance at the daily, monthly and yearly scales, with CV RMSE values ranging from 9.0 to 15.1 μg/m$^3$ for the $PM_{2.5}$ concentration and from 19.1 to 28.8 μg/m$^3$ for the $PM_{10}$ concentration.

The quality of the $PM_{2.5}$ and $PM_{10}$ reanalysis data in the different regions of China is further summarized in Table S1-2. At the hourly scale, small negative biases of the $PM_{2.5}$ reanalysis data were found in the NCP (-4.8%), NE (-5.8%), SE (-3.8%) and SW (-3.4%) regions. The biases in the NW and central regions were relatively large, with CV normalized mean bias (CV NMB) values of approximately -7.3% and -8.2%, respectively. Two reasons might explain the large biases in these two regions. First, the observation sites are sparse in the NW and central regions. As a result, the $PM_{2.5}$ concentration is not suitably constrained at certain sites in the CV method. Second, the emissions of $PM_{2.5}$ and its precursors might be very low in these two regions, leading to underestimation of the background errors since we only considered the emission uncertainty in the ensemble simulations. Although this problem was alleviated by using the inflation technique to compensate for the missing errors, the overconfident model results still degraded the assimilation performance to a certain extent, making the analysis less influenced by the observations. The errors of the $PM_{2.5}$ reanalysis data exhibited apparent spatial differences (Table S1). The CV RMSE values were the smallest in the SE (14.9 μg/m$^3$) and SW (16.5 μg/m$^3$) regions and increased to ~25 μg/m$^3$ in the NCP, NE and central regions. Consistent with the bias distributions, the largest CV RMSE value was found in the NW region, which reached 52.1 μg/m$^3$ but was still much smaller than the RMSE value of the base simulation (73.0 μg/m$^3$). The errors of the $PM_{2.5}$ reanalysis data were small at the daily, monthly and yearly scales, with CV RMSE values of approximately 10.6– 39.4 μg/m$^3$ at the daily scale, 7.4–26.9 μg/m$^3$ at the monthly scale and 6.1–23.5 μg/m$^3$ at the yearly scale. In terms of the hourly $PM_{10}$ reanalysis data, the CV results (Table S2) indicated that small negative biases occurred in the NCP, NE, SE and

SW regions, ranging from -9.6% (NE region) to -5.9% (SE region). The biases were larger in the NW and central regions, with
the CV NBM values increasing to approximately 18.0% and 14.1%, respectively. The errors of the $PM_{10}$ reanalysis data also
exhibited a spatial heterogeneity. The CV RMSE value was the smallest in the SE (26.0 μg/m³) and SW (30.2 μg/m³) regions
and increased to approximately 39.8 and 43.7 μg/m³ in the NE and NCP regions, respectively. The largest errors were found
in the central and NW regions, with CV RMSE values of approximately 105.5 and 57.3 μg/m³, respectively. The $PM_{10}$
reanalysis data revealed small errors at the daily, monthly and yearly scales, with CV RMSE values of approximately 18.6–
85.5 μg/m³ at the daily scale, 13.7–64.0 μg/m³ at the monthly scale and 12.3–55.8 μg/m³ at the yearly scale.
**4.1.3 Trend study of the PM reanalysis data over China**
A realistic representation of the observed interannual change is another important aspect of the reanalysis dataset. The
performance of the reanalysis data in representing the observed interannual changes in the $PM_{2.5}$ and $PM_{10}$ concentrations was
thus evaluated nationwide and in the different regions of China. Figures 7–8 show time series of the monthly mean $PM_{2.5}$ and
$PM_{10}$ concentrations nationwide and in the different regions. The observed national $PM_{2.5}$ concentration revealed a profound
seasonal cycle with the highest concentration in winter and the lowest level in summer. The annual trends of the $PM_{2.5}$ and
$PM_{10}$ concentrations were also calculated using the Mann-Kendall (M-K) trend test and the Theil-Sen trend estimation method,
which are summarized in Table 3. A significant negative trend was observed in the $PM_{2.5}$ concentration nationwide, with a
calculated annual trend of approximately -5.8 (p<0.05) $μg \cdot m^{-3} \cdot yr^{-1}$. The NE and NCP regions exhibited the highest
negative trends among the six regions, with calculated trends of approximately -7.5 (p<0.05) and -7.0 (p<0.05) $μg \cdot m^{-3} \cdot yr^{-1}$,
respectively. In the other regions, the negative trends ranged from -6.3 to -5.2 $μg \cdot m^{-3} \cdot yr^{-1}$. The base simulation suitably
reproduced the observed seasonal cycle of the $PM_{2.5}$ concentration in all regions. The magnitude of the $PM_{2.5}$ concentration in
2013 was also captured well in the different regions, suggesting that the emission inventories of 2010 were generally reasonable
for the simulation of the $PM_{2.5}$ concentration in 2013. However, starting from 2014, the base simulation tended to overestimate
the observations in the NCP, SE and SW regions, indicating that the emission inventory of 2010 may be too high for the
simulation of the $PM_{2.5}$ concentration in these regions after 2014. In contrast, the base simulation significantly underestimated
the $PM_{2.5}$ concentration in the NW region. The model performance of the base simulation was relatively good in the NE and
central regions throughout the six years. Although the base simulation captured the negative trends of the observed $PM_{2.5}$
concentration in China and the different regions, the simulated trends were much lower than those indicated by the observations.
Since we adopted the same emission inventory in the simulations of the air pollutants in the different years, the simulated
trends in the base simulation were only driven by the variations in meteorological conditions. This suggests that the change in
meteorological conditions only explained a small proportion of the negative trends in the $PM_{2.5}$ concentration in China and
that emission reductions contributed more to the decline in the $PM_{2.5}$ concentration. The CV run agreed better with the
observations. The observed trends of the $PM_{2.5}$ concentration in China and each subregion were all suitably captured by the
reanalysis in the CV run. Similar results were obtained for the analysis of the trend of the $PM_{10}$ concentration, as shown in Fig.
8. The observed $PM_{10}$ concentration also exhibited significant negative trends, which were captured well by the $PM_{10}$ reanalysis
in the CV run. The base simulation attained a better performance in reproducing the $PM_{10}$ concentration in China than in
reproducing the $PM_{2.5}$ concentration, while significant underestimations of the $PM_{10}$ concentration occurred in the NW and
central regions. The calculated negative trends of the base simulation were still lower than those indicated by the observations.
This again highlights the large contributions of emission reduction to the improvement of the air quality in China in these years.

### 481 4.1.4 Independent validation of the $PM_{2.5}$ reanalysis data

In addition to the CV method, the $PM_{2.5}$ reanalysis data were further validated against an independent dataset acquired
from the U.S. Department State Air Quality Monitoring Program over China (http://www.stateair.net/; last accessed: 17 April
2020), which contains the hourly $PM_{2.5}$ concentration in Beijing, Chengdu, Guangzhou, Shanghai and Shenyang cities. Table
4 presents a comparison of the observed $PM_{2.5}$ concentrations to those obtained from the CAQRA dataset and base simulation.
The results indicated that the magnitude and variability of the $PM_{2.5}$ reanalysis data agreed better with those of the observed
$PM_{2.5}$ concentrations in all cities. Both the MBE and RMSE values were greatly reduced in the CAQRA dataset, which only
ranged from -7.1 to -0.3 $\mu g \cdot m^{-3}$ and from 16.8 to 33.6 $\mu g \cdot m^{-3}$, respectively, in these cities. The correlation coefficient was
also greatly improved in CAQRA ($R^2 = 0.74–0.86$) over the base simulation ($R^2 = 0.09–0.38$). These results confirm that the
CAQRA dataset attains a high quality performance in representing the $PM_{2.5}$ pollution in China in these years.

### 491 4.1.5 Comparison to the satellite-estimated $PM_{2.5}$ concentration

Previous studies have shown that estimating the ground-based $PM_{2.5}$ concentration from the satellite-derived AOD is an
effective way to map the $PM_{2.5}$ concentration with a good accuracy. To further demonstrate the accuracy of our $PM_{2.5}$ reanalysis
data, we also compared the accuracy to that of satellite-estimated $PM_{2.5}$ concentrations. Table 5 summarizes several
representative studies focusing on the estimation of the ground-based $PM_{2.5}$ concentration in China at the national level using
different kinds of methods. Most of these studies estimated the ground-based $PM_{2.5}$ concentration at the daily scale since they
employed polar-orbiting satellite data (e.g., MODIS) that only provide daily AOD observations. The estimation conducted by
Liu et al. (2019) was an exception which exhibited an hourly resolution due to the use of AOD measurements from a
geostationary satellite (Himawari-8). The horizontal resolution in these studies was mainly approximately 10 km except that
of Lin et al. (2018), which revealed the finest horizontal resolution (1 km), and that of Zhan et al., 2017, which revealed the
coarsest horizontal resolution (0.5°). Few studies have provided long-term $PM_{2.5}$ data covering recent years. In comparison,
our $PM_{2.5}$ reanalysis data provide long-term data in China at a fine temporal resolution (1 h) and a high accuracy. A fine
temporal resolution is important for epidemiological studies, especially for the assessment of the acute health effects of air
pollution. Furthermore, the accuracy of our reanalysis data (CV $R^2 = 0.86$ and CV RMSE = 15.1 $\mu g \cdot m^{-3}$) was also higher
than that of most of these satellite estimates (CV $R^2 = 0.56–0.86$ and CV RMSE = 15.0–20.2 $\mu g \cdot m^{-3}$).

## 4.2 Gases

### 4.2.1 Spatial distribution of the reanalysis data of gaseous air pollutants over China

Next, we present the reanalysis fields for gaseous air pollutants in China, namely, $SO_2$, CO, $NO_2$ and $O_3$. Figure 9 shows the spatial distribution of the six-year average $SO_2$ and CO concentrations in China obtained from the CAQRA dataset, base simulation and observations. The $SO_2$ reanalysis data captured the magnitude and spatial distribution of the $SO_2$ concentration in China well, while the base simulation greatly overestimated the $SO_2$ concentration due to the positive biases of the $SO_2$ emissions in the simulations. Consistent with the observations, the $SO_2$ reanalysis data exhibited high spatial heterogeneity, with the highest values located in the NCP region, especially in Shandong, Shanxi and Hebei provinces. Several $SO_2$ concentration hotspots were also found in the NE region. $SO_2$ is mainly emitted from fossil fuel consumption, especially coal burning (Lu et al., 2010). Shandong, Shanxi, Inner Mongolia and Hebei provinces are the four largest consumers of coal in China according to the China Energy Statistical Yearbook (NBSC 2017a, b), which explains the high $SO_2$ concentrations in these provinces. The spatial distribution of the CO reanalysis data was similar to that of the $SO_2$ reanalysis data and agreed well with the observed spatial distribution. In contrast, the base simulation highly underestimated the CO concentration, especially in the NCP region. In addition, both the observations and reanalysis data showed CO concentration hotspots in the NW region and Xizang Province, while these hotspots were largely underestimated or even missing in the base simulation. According to previous studies, such underestimation might be related to underestimated CO emissions in China (Kong et al., 2020; Tang et al., 2013). In regard to $NO_2$ (Fig. 10), both the reanalysis data and base simulation captured the observed magnitude and spatial distribution of the $NO_2$ concentration in China. High $NO_2$ concentrations generally occurred in the NCP region and the major city clusters in China. However, the base simulation generally revealed an underestimated $NO_2$ concentration in China. The spatial distribution of the $O_3$ concentration (Fig. 10) demonstrated a lower spatial heterogeneity than that of the other gases. The $O_3$ reanalysis data suitably captured the observed magnitude and spatial distribution of the $O_3$ concentration in China, while the base simulation generally underestimated the $O_3$ concentration in China. Figures S9–12 further show seasonal maps of the reanalysis fields of these gases. All gases exhibited a profound seasonal cycle, with maximum values observed in winter and the lowest values in summer except $O_3$, which demonstrated the opposite seasonal cycle. The highest $SO_2$, CO and $NO_2$ concentrations in winter could occur due to the increased anthropogenic emissions and the more stable atmospheric conditions during this season. Regarding $O_3$, the highest value in summer was closely related to the enhanced photochemical reactions in summer associated with the high temperature and solar radiance.

### 4.2.2 Assessment of the gas reanalysis data over China

Evaluation results of the above gas reanalysis data are provided in Table 2. The table indicates that the reanalysis data attain an excellent performance in representing the magnitude and variability of these gaseous air pollutants in China, with CV $R^2$ values ranging from 0.51 for $SO_2$ to 0.76 for $O_3$ and CV MBE (CV NMB) values of approximately -2.0 $\mu g \cdot m^{-3}$ (-8.5%), -2.3 $\mu g \cdot m^{-3}$ (-6.9%), -0.06 $mg \cdot m^{-3}$ (-6.1%) and -2.3 $\mu g \cdot m^{-3}$ (-4.0%) for the hourly $SO_2$, $NO_2$, CO and $O_3$ reanalysis data,

respectively. Compared to the base simulation, the errors were reduced by approximately half in the reanalysis data with CV RMSE values of approximately 24.9 $\mu g \cdot m^{-3}$, 16.4 $\mu g \cdot m^{-3}$, 0.54 $mg \cdot m^{-3}$ and 21.9 $\mu g \cdot m^{-3}$ for the hourly $SO_2$, $NO_2$, CO and $O_3$ reanalysis data, respectively. The reanalysis data achieved a good performance at the daily, monthly and yearly scales. The CV RMSE values of the daily $SO_2$ and $NO_2$ reanalysis data were also smaller than those of the $SO_2$ and $NO_2$ concentration datasets in China previously developed by Zhan et al. (2018) and Zhang et al. (2019), respectively, based on the random-forest-spatiotemporal-kriging model wherein the RMSE values of the daily $SO_2$ and $NO_2$ concentrations were estimated to be 19.5 and 13.3 $\mu g \cdot m^{-3}$, respectively.

In terms of the different regions (Tables S3–6), the hourly $SO_2$ reanalysis data indicated small negative biases (approximately 2–10%) in all regions except in the central region, where the negative bias was relatively large (17.0%). The smallest CV RMSE values of the $SO_2$ reanalysis data were observed in the SE, SW and NW regions (smaller than 25 $\mu g \cdot m^{-3}$), while in the other regions, the CV RMSE values exceeded 30 $\mu g \cdot m^{-3}$. The hourly $NO_2$ reanalysis data showed small negative biases in all regions, which were relatively small in the NE, NCP and SE regions (ranging from -5.9 to -3.5%) and were relatively large in the SW, NW and central regions (ranging from -15.1 to -12.9%). The CV RMSE for the hourly $NO_2$ reanalysis data was approximately 15 $\mu g \cdot m^{-3}$ in all regions except in the NW (24.3 $\mu g \cdot m^{-3}$) and central (20.5 $\mu g \cdot m^{-3}$) regions. The hourly CO reanalysis data exhibited small negative biases in all regions. The largest biases were still found in the NW region, which reached approximately 15.0%, while in the other regions, the biases ranged from -11.2% to -2.5%. The CV RMSE values for the hourly CO reanalysis data were the smallest in South China (approximately 0.39 and 0.46 $mg \cdot m^{-3}$ in the SE and SW regions, respectively) and increased to 0.64 and 0.59 $mg \cdot m^{-3}$ in the NCP and NE regions, respectively. The largest CV RMSE was observed in the NW region, which amounted to approximately 1.13 $mg \cdot m^{-3}$. The biases of the hourly $O_3$ reanalysis data were uniformly distributed in the different regions, with the CV NMB value ranging from -6.1% to 1.4%. Similarly, the CV RMSE value of the $O_3$ reanalysis data was approximately 20 $\mu g \cdot m^{-3}$ in all regions except in the NW region (28.3 $\mu g \cdot m^{-3}$).

**4.2.3 Trend study of the gas reanalysis data over China**

Figure 11 shows time series of the monthly mean $SO_2$ concentration in China obtained from the CV run, base simulation and observations. Additionally, time series of the monthly mean $SO_2$ concentration in the different regions are shown. The observed $SO_2$ concentrations showed significant negative trends (P<0.05) in China (-6.2 $\mu g \cdot m^{-3} \cdot yr^{-1}$, Table 6) and in all regions (ranging from -2.3 to -9.5 $\mu g \cdot m^{-3} \cdot yr^{-1}$, Table 6) due to the large reductions in $SO_2$ emissions across China. During the $11^{th}$-$13^{rd}$ Five-Year Plans (FYPs) and the Air Pollution Prevention and Control Plan, the Chinese government invested great efforts to reduce $SO_2$ emissions, such as the installation of flue-gas desulfurization (FGD) and selective catalytic reduction systems, construction of large units, decommissioning of small units and replacement of coal with cleaner energies (Li et al., 2017; Zheng et al., 2018b). As a result, the $SO_2$ emissions substantially decreased in China, especially in the industrial and power sectors. The base simulation significantly overestimated the $SO_2$ concentration in all regions, especially after 2013.

The negative trends of the $SO_2$ concentration were also largely underestimated in the base simulation. In contrast, the $SO_2$
reanalysis data captured the magnitude and negative trends of the observed $SO_2$ concentrations in China and in all regions well.
The $NO_2$ observations showed negative trends in China as well (Fig. 12). However, the negative trend was not significant
except in the NE region (Table 6). This is consistent with the small reductions in $NO_x$ emissions (21%) in China due to the
small changes in the emissions originating from the transportation sector, accounting for almost one-third of the $NO_x$ emissions
in China. The pollution controls applied in the transportation section were exactly offset by the growing emissions related to
vehicle growth (Zheng et al., 2018b). The base simulation generally underestimated the $NO_2$ concentration during the
wintertime, and the observed negative trends of the $NO_2$ concentration were also underestimated in all regions. By assimilating
the observed $NO_2$ concentrations, the reanalysis data agreed better with the observations both in regard to the magnitude and
negative trends. The CO observations exhibited significant negative trends in all regions except in the NW region (Fig. 13),
with calculated negative trends ranging from -0.18 to -0.06 $\mu g \cdot m^{-3} \cdot yr^{-1}$. Such negative trends have also been observed in
satellite measurements, such as MOPITT observations (Zheng et al., 2018a), which are mainly attributed to the reduced
anthropogenic emissions in China, as suggested by both bottom-up and top-down methods (Zheng et al., 2019). The base
simulation largely underestimated the CO concentration in all regions. In addition, the negative trends of the CO concentration
were also notably underestimated in the base simulation, which highlights the major contribution of emission reduction to the
decreased CO concentration in these regions. The CO reanalysis data agreed well with the observations and captured the
negative trends of the CO concentration in all regions. The $O_3$ concentration exhibited the opposite trend to that exhibited by
the other air pollutants (Fig. 14), which revealed significant positive trends in all regions, ranging from 2.3 to 5.4 $\mu g \cdot m^{-3} \cdot$
$yr^{-1}$ and indicating enhanced photochemical pollution in China. This phenomenon has been observed and investigated by Li
et al. (2019), who suggested that the rapid decrease in the $PM_{2.5}$ concentration and the resultant reduction in the aerosol sink
of hydroperoxyl ($HO_2$) radicals were important factors contributing to the enhanced $O_3$ concentration in China. The base
simulation generally captured the magnitude of the $O_3$ concentration in the SE, SW, NW and central regions but underestimated
the $O_3$ concentration in the NCP and NE regions, especially in spring and summer. In addition, the base simulation
underestimated the observed positive trends of the $O_3$ concentration in all regions, which suggests that meteorological
variability only contributed a small proportion of the observed $O_3$ trend in China. Again, the $O_3$ reanalysis data are substantially
better than the base simulation and suitably reproduce the observed trends of the $O_3$ concentration in all regions.
**4.2.4 Comparison to the CAMS reanalysis data**
To further evaluate the accuracy of our reanalysis dataset for gaseous air pollutants, the CAMSRA dataset produced by
the ECMWF (Inness et al., 2019) was employed as a reference in a comparison to our reanalysis dataset. The CAMSRA dataset
is the latest global reanalysis dataset of the atmospheric composition, which assimilates satellite retrievals of $O_3$, CO, $NO_2$ and
AOD. Three-hour reanalysis data of the $SO_2$, $NO_2$, CO and $O_3$ concentrations at the surface model level from 2013 to 2018
were adopted in this study, which were downloaded from https://atmosphere.copernicus.eu/copernicus-releases-new-global-
reanalysis-data-set-atmospheric-composition (last accessed: 17 April 2020) at a resolution of 1 degree by 1 degree. Here, we
only focus on a comparison of the gaseous pollutants since the CAMSRA dataset does not provide $PM_{2.5}$ and $PM_{10}$
concentrations.
Figure 15 shows the spatial distribution of the six-year average concentration of these gaseous air pollutants in China
obtained from the CAMSRA dataset. Compared to the spatial distributions determined with the CAQRA dataset and
observations (Figs. 9–10), the CAMSRA dataset greatly overestimates the surface $SO_2$ and $O_3$ concentrations in China. In
addition, due to the higher spatial resolution (15 km) of the CAQRA dataset than that of the CAMSRA dataset (approximately
50 km), our products provide more detailed spatial patterns of the surface air pollutants in China, which are better suited for
air quality studies at the regional scale. Table 7 quantitatively compares the accuracy of the CAQRA dataset to that of the
CAMSRA dataset in the estimation of the surface concentrations of gaseous air pollutants in China. Compared to CAMSRA
($R^2$ = 0.00–0.23), CAQRA attains a much better performance in capturing the spatiotemporal variability in the surface
concentrations of gaseous air pollutants in China, with $R^2$ values ranging from 0.53 to 0.77. The MBE and RMSE values are
also smaller in the CAQRA dataset than those in the CAMSRA dataset, especially for the $SO_2$ and $O_3$ concentrations. This is
attributed to the assimilation of surface observations in CAQRA, while CAMSRA only assimilates satellite retrievals. These
results suggest that the CAQRA dataset provides surface air quality datasets in China of a higher quality than the air quality
datasets provided by the CAMSRA dataset, which is especially valuable for future relevant studies with high demands in
spatiotemporal resolution and accuracy.
**5 Conclusions**
A high-resolution CAQRA dataset was produced in this study by assimilating surface observations of the $PM_{2.5}$, $PM_{10}$,
$SO_2$, $NO_2$, CO and $O_3$ concentrations retrieved from the CNEMC. This dataset provides time-consistent concentration fields
of $PM_{2.5}$, $PM_{10}$, $SO_2$, $NO_2$, CO and $O_3$ in China from 2013 to 2018 (will be extended in the future on a yearly basis) at high
spatial (15 km) and temporal (1 hour) resolutions. The CAQRA dataset was produced with the ChemDAS, which applied the
NAQPMS model as the forecast model, and the LETKF to assimilate the observations in the postprocessing mode. The
background error covariance was calculated from ensemble simulations, which considered the emission uncertainties of the
major air pollutants. An inflation technique was also applied to dynamically inflate the background error to prevent
underestimation of the true background error covariance.
The fivefold CV method was employed to validate the reanalysis dataset, which provided us with the first indication of
the quality of the CAQRA dataset. The validation results suggested that the CAQRA dataset attains an excellent performance
in representing the spatiotemporal variability of surface air pollutants in China, with CV $R^2$ values ranging from 0.52 for the
hourly $SO_2$ concentration to 0.81 for the hourly $PM_{2.5}$ concentration. The CV MBE values of the reanalysis data were -2.6 µg ·
$m^{-3}$, -6.8 µg · $m^{-3}$, -2.0 µg · $m^{-3}$, -2.3 µg · $m^{-3}$, -0.06 mg · $m^{-3}$ and -2.3 µg · $m^{-3}$ for the hourly concentrations of $PM_{2.5}$,
$PM_{10}$, $SO_2$, $NO_2$, CO and $O_3$, respectively. The CV RMSE values of the reanalysis data for these air pollutants were estimated
to be approximately 21.3 µg · $m^{-3}$, 39.3 µg · $m^{-3}$, 24.9 µg · $m^{-3}$, 16.4 µg · $m^{-3}$, 0.54 mg · $m^{-3}$ and 21.9 µg · $m^{-3}$,
respectively. In the different regions of China, the NW and central regions exhibited relatively large biases and errors, which
mainly occurred due to the relatively sparse observations and underestimated background errors. The Chinese air quality has
substantially changed over the last six years. The observations indicate significant decreasing trends for all air pollutants except
$O_3$, which shows an increasing trend over the last six years. The reanalysis data reveal an excellent performance in representing
the trends of all air pollutants in China, suggesting the suitability of the reanalysis data for air pollutant trend analysis in China.
In addition to the CV method, the $PM_{2.5}$ reanalysis data were also evaluated against independent observations retrieved
from the U.S. Department State Air Quality Monitoring Program over China. The results suggested that the reanalysis data
suitably reproduce the magnitude and variability of the observed $PM_{2.5}$ concentration in all cities, with the MBE and RMSE
values only ranging from -7.1 to -0.3 $\mu g \cdot m^{-3}$ and from 16.8 to 33.6 $\mu g \cdot m^{-3}$, respectively. The reanalysis data of the gaseous
air pollutants were also compared to the latest global reanalysis data contained in the CAMSRA dataset produced by the
ECMWF. The CAMSRA dataset is of great value in providing three-dimensional distributions of multiple chemical species
globally. As a regional dataset, our products attain a higher spatial resolution than does the CAMSRA dataset, which could
better suit air quality studies at the regional scale. Although our products only provide the surface concentrations of six
conventional air pollutants in China, the accuracy of the CAQRA dataset was estimated to be higher than that of the CAMSRA
dataset due to the assimilation of surface observations. Hence, our products exhibit their own value in regional air quality
studies with high demands in spatiotemporal resolution and accuracy. We also compared our $PM_{2.5}$ reanalysis data to previous
satellite estimates of the surface $PM_{2.5}$ concentration, which revealed that the $PM_{2.5}$ reanalysis data are more accurate than
most satellite estimates and exhibit a relatively fine temporal resolution.
As the first version of the CAQRA dataset, certain limitations remain that potential users should be aware of. First, the
discontinuities in the availability and coverage of assimilated observations will affect the reanalysis quality and the estimated
interannual trends. As shown in Sect.3.1, there has been a consistent increase in the number of assimilated observations from
2013 to 2015 due to the increases of observation sites. The smaller number of assimilated observations in 2013 and 2014 would
provide less constrains on the background state and thus degrade the reanalysis in these two years. This may cause spurious
interannual changes and trends from 2013 to 2018. Thus, cautions are needed when using the reanalysis for long-term air
quality change from 2013 to 2018. However, this problem would be not serious after 2015 when the number of assimilated
observations become stable. In addition, the observation sites used in the assimilation are mainly urban or suburban sites that
do not provide enough information on the air pollution in rural areas, which may influence the quality of CAQRA in rural
areas. Secondly, we only perturbed the emissions to represent the forecast uncertainty in this study, which may underestimate
the forecast uncertainty due to the omitting of other error sources, such as the uncertainty in poorly parameterized physical or
chemical processes, and the uncertainty in meteorological simulation. The limited ensemble size would also lead to
underestimation of the forecast error especially in the high-resolution assimilation applications. Although the inflation method
is used to compensate for the missing errors, the underestimated forecast uncertainty would still degrad the assimilation
performance to a certain extent as exemplified by the larger biases in the reanalysis over NW and Central regions. Thirdly, we
did not consider the annual trend of emissions in the ensemble simulation. This would lead to temporal changes in the statistics

of innovation due to the substantial changes of observations, which would influence the long stability of the data assimilation as suggested by the $\chi^2$ test although the OmA statistics generally confirms a passable stability in our assimilation system. Last but not least, the current CAQRA only contains the surface concentrations of the air pollutants in China which cannot provide the information on the vertical structure of the air pollutants. to further improve the accuracy of our air quality reanalysis dataset, in the future, an online EnKF run could be conducted to simultaneously correct the emissions and concentrations. More observation types, such as observation data of the $PM_{2.5}$ composition, could also be assimilated to provide $PM_{2.5}$ composition fields in China, which could support both epidemiological studies and climate research.

## Data availability

The whole CAQRA reanalysis dataset can be freely downloaded at https://doi.org/10.11922/sciencedb.00053 (Tang et al., 2020a), and the prototype product, which contains the monthly and annual means of the CAQRA dataset, is available at https://doi.org/10.11922/sciencedb.00092 (Tang et al., 2020b).

## Author contributions

X.T., J.Z., and Z.W. conceived and designed the project; H.W., L.K., X.T., and L.W. established the data assimilation system; Q.W. and L.K. performed the meteorology simulations; X.T., L.K., H.C., H.W., H.Z., G.J. and M.L. conducted the ensemble simulations with the NAQPMS model; J.L., L.Z., W.W., B.L., Q.W., D.C. and T.S. provided the air quality monitoring data; W.H. executed the quality control of the observation data; F.L. estimated the representativeness error of the observations; and L.K. carried out the CAQRA calculations, generated the figures and wrote the paper with comments provided by G.C.

## Competing interests

The authors declare that they have no conflicts of interest.

## Acknowledgements

This study was funded by the National Natural Science Foundation (Grant Nos. 91644216, 41575128, and 41875164), the CAS Strategic Priority Research Program (Grant No. XDA19040201), and the CAS Information Technology Program (Grant No. XXH13506-302).

 **Tables**

 **Table 1: Uncertainties in the emissions of the different species**

| Species | $SO_2$[a] | $NO_x$[a] | CO[a] | Non-methane volatile organic compounds (NMVOCs)[a] | $NH_3$[b] | $PM_{10}$[a] | $PM_{2.5}$[a] | Black carbon (BC)[a] | Organic carbon (OC)[a] |
|---|---|---|---|---|---|---|---|---|---|
| Emission Uncertainty | 12% | 31% | 70% | 68% | 53% | 132% | 130% | 208% | 258% |

[a] Emission uncertainty obtained from Zhang et al. (2009)
[b] Emission uncertainty obtained from Streets et al. (2003)



















**Table 2: Site-based cross-validation results for the reanalysis data (outside brackets) and base simulation (inside brackets) from 2013 to 2018 at the different temporal scales**

| | PM$_{2.5}$ ($\mu g/m^3$) | | | | PM$_{10}$ ($\mu g/m^3$) | | | |
|---|---|---|---|---|---|---|---|---|
| | $R^2$ | MBE | NMB (%) | RMSE | $R^2$ | MBE | NMB (%) | RMSE |
| Hourly | 0.81 (0.26) | -2.6 (17.6) | -4.9 (34.7) | 21.3 (54.1) | 0.72 (0.17) | -6.8 (-7.6) | -7.8 (-8.7) | 39.3 (75.7) |
| Daily | 0.86 (0.32) | -2.5 (17.4) | -4.9 (34.3) | 15.1 (46.4) | 0.81 (0.22) | -6.7 (-7.0) | -7.7 (-8.1) | 28.8 (64.1) |
| Monthly | 0.88 (0.40) | -2.5 (17.4) | -5.0 (34.1) | 10.3 (33.6) | 0.83 (0.28) | -6.7 (-7.3) | -7.7 (-8.4) | 21.1 (44.4) |
| Yearly | 0.86 (0.37) | -3.0 (15.2) | -5.6 (28.7) | 9.0 (28.9) | 0.79 (0.27) | -7.5 (-10.2) | -8.3 (-11.3) | 19.1 (38.2) |

| | SO$_2$ ($\mu g/m^3$) | | | | NO$_2$ ($\mu g/m^3$) | | | |
|---|---|---|---|---|---|---|---|---|
| | $R^2$ | MBE | NMB (%) | RMSE | $R^2$ | MBE | NMB (%) | RMSE |
| Hourly | 0.52 (0.03) | -2.0 (25.5) | -8.5 (106.6) | 24.9 (67.2) | 0.61 (0.22) | -2.3 (-5.0) | -6.9 (-14.8) | 16.4 (24.9) |
| Daily | 0.67 (0.04) | -2.0 (25.6) | -8.5 (106.9) | 17.5 (59.3) | 0.67 (0.27) | -2.3 (-5.0) | -6.8 (-14.8) | 12.3 (19.9) |
| Monthly | 0.74 (0.04) | -2.1 (25.4) | -8.6 (105.7) | 13.2 (52.0) | 0.67 (0.34) | -2.3 (-5.0) | -6.8 (-14.8) | 10.0 (15.9) |
| Yearly | 0.71 (0.04) | -2.6 (23.1) | -9.9 (87.2) | 12.0 (47.5) | 0.62 (0.42) | -2.5 (-5.9) | -7.3 (-17.3) | 9.1 (13.6) |

| | CO ($mg/m^3$) | | | | O$_3$ ($\mu g/m^3$) | | | |
|---|---|---|---|---|---|---|---|---|
| | $R^2$ | MBE | NMB (%) | RMSE | $R^2$ | MBE | NMB (%) | RMSE |
| Hourly | 0.55 (0.17) | -0.06 (-0.47) | -6.1 (-44.7) | 0.54 (0.87) | 0.76 (0.35) | -2.3 (-10.5) | -4.0 (-17.8) | 21.9 (38.3) |
| Daily | 0.61 (0.20) | -0.06 (-0.47) | -5.8 (-44.6) | 0.44 (0.77) | 0.74 (0.25) | -2.3 (-10.4) | -3.9 (-17.8) | 16.6 (31.3) |
| Monthly | 0.62 (0.21) | -0.06 (-0.47) | -6.0 (-44.7) | 0.36 (0.69) | 0.74 (0.28) | -2.3 (-10.4) | -3.9 (-17.8) | 13.1 (25.3) |
| Yearly | 0.52 (0.09) | -0.08 (-0.51) | -6.9 (-46.7) | 0.37 (0.72) | 0.53 (0.03) | -2.2 (-9.8) | -3.8 (-17.2) | 10.4 (21.2) |

**Table 3: Calculated annual trends of the PM$_{2.5}$ and PM$_{10}$ concentrations in China**

| | PM$_{2.5}$ (µg/m$^3$) | | | PM$_{10}$ (µg/m$^3$) | | |
|---|---|---|---|---|---|---|
| | Observation | Cross-validation | Base simulation | Observation | Cross-validation | Base simulation |
| China | **-5.8 (-13.4, -3.5)[a]** | **-5.0 (-12.6, -3.1)** | **-2.0 (-3.6, -0.7)** | **-7.2 (-18.4, -3.2)** | **-6.0 (-17.0, -2.9)** | **-2.5 (-3.6, -0.7)** |
| NCP | **-7.0 (-15.7, -5.5)** | **-6.6 (-14.5, -4.8)** | **-3.5 (-4.7, -1.9)** | **-8.3 (-20.4, -5.1)** | **-7.6 (-19.2, -4.4)** | **-4.2 (-4.7, -1.9)** |
| NE | **-7.5 (-11.0, -3.9)** | **-6.7 (-10.0, -3.5)** | **-3.2 (-5.8, -1.2)** | **-11.2 (-17.4, -4.7)** | **-10.4 (-16.4, -4.7)** | **-3.7 (-5.8, -1.2)** |
| SE | **-5.2 (-11.3, -2.8)** | **-4.9 (-10.6, -2.7)** | -0.9 (-3.1, 1.3) | **-6.0 (-14.9, -2.4)** | **-5.8 (-13.2, -1.9)** | -1.6 (-3.1, 1.3) |
| SW | **-6.3 (-12.8, -2.6)** | **-4.9 (-12.2, -2.4)** | -1.4 (-7.5, 0.4) | **-7.9 (-19.9, -2.2)** | **-5.5 (-17.5, -2.1)** | -1.3 (-7.5, 0.4) |
| NW | -5.7 (-11.6, 2.1)[b] | -3.3 (-10.7, 1.8) | -1.3 (-4.9, 2.9) | -0.5 (-14.4, 1.6) | -2.2 (-8.5, 3.4) | -2.3 (-4.9, 2.9) |
| Central | **-5.8 (-19.8, -0.8)** | -3.6 (-17.7, 0.2) | -0.6 (-5.9, 0.9) | **-8.9 (-28.5, 0.2)** | -6.8 (-26.9, 0.5) | -2.0 (-5.9, 0.9) |

[a] The bold font denotes that the calculated trend is significant at the 0.05 significance level, and the values in brackets denote
the 95% confidence interval.





**Table 4: Independent validation results of the CAQRA dataset (outside brackets) and base simulation (inside brackets)**
**against the observation data retrieved from the U.S. Department State Air Quality Monitoring Program over China**

| | R$^2$ | MBE (µg/m$^3$) | NMB (%) | RMSE (µg/m$^3$) |
|---|---|---|---|---|
| Beijing | 0.86 (0.37) | -0.3 (11.4) | -0.3 (13.2) | 33.6 (75.6) |
| Shanghai | 0.86 (0.34) | 5.5 (39.6) | 10.9 (78.3) | 17.1 (64.8) |
| Chengdu | 0.85 (0.19) | -7.1 (59.3) | -8.9 (74.7) | 23.1 (91.5) |
| Guangzhou | 0.74 (0.09) | -3.3 (11.1) | -7.5 (25.1) | 16.8 (38.8) |
| Shenyang | 0.85 (0.29) | -2.2 (16.8) | -3.2 (24.3) | 24.8 (59.1) |




**Table 5 Comparison of the accuracy of our PM$_{2.5}$ reanalysis data to that of satellite estimates**

| Reference | Spatial resolution | Temporal resolution | Temporal coverage | CV R$^2$ | CV RMSE | Method |
|---|---|---|---|---|---|---|
| Ma et al. (2016) | 0.1° ×0.1° | daily | 2004–2013 | 0.79 | 27.4 | LME + GAM |
| Xue et al. (2019) | 0.1° ×0.1° | daily | 2000–2016 | 0.56 | 30.2 | CTM + HD-expansion + GAM |
| Xue et al. (2017) | 0.1° ×0.1° | daily | 2014 | 0.72 | 23.0 | CTM + LME + spatiotemporal kriging |
| Chen et al. (2018) | 0.1° ×0.1° | daily | 2005–2016 | 0.83 | 18.1 | RF |
| Lin et al. (2018) | 1 km× 1km | daily | 2001 – 2015 | 0.78[a] | 19.3[a] | Semi-empirical |
| Chen et al. (2019) | 3 km×3 km | daily | 2014 – 2015 | 0.86 | 15.0 | XGBoost + NELRM |
| Yao et al. (2019) | 6 km×6 km | daily | 2014 | 0.60 | 21.8 | TEFR + GWR |
| You et al. (2016) | 0.1° ×0.1° | daily | 2014 | 0.79 | 18.6 | GWR |
| Zhan et al. (2017) | 0.5° ×0.5° | daily | 2014 | 0.76 | 23.0 | GW-GBM |
| Li et al. (2017b) | 0.1° ×0.1° | daily | 2015 | 0.82 | 16.4 | Geoi-DBN |
| Liu et al. (2019) | 0.125° ×0.125° | hourly | 2016 | 0.86 | 17.3 | RF |
| This study | 15 km× 15km | hourly | 2013–2018 | 0.81 | 21.3 | EnKF |
|  |  | daily | 2013–2018 | 0.86 | 15.1 | EnKF |

[a] The accuracy of the PM$_{2.5}$ estimates of Lin et al. (2018) was assessed at the monthly scale.
LME: Linear mixed-effect model
GWR: Geographically weighted regression model
GAM: Generalized additive model
HD-expansion: High-dimensional expansion
RF: Random forest
XGBoost: Extreme gradient boosting
NELRM: Non-linear exposure-lag-response model
TEFR: Time fixed-effects regression model
GW-GBM: Geographically weighted gradient boosting machine
Geoi-DBN: Geographical deep belief network

**Table 6: Calculated annual trends of the SO₂, NO₂, CO and O₃ concentrations in China**

| | SO₂ (μg/m³) | | | NO₂ (μg/m³) | | |
|---|---|---|---|---|---|---|
| | Observation | Cross-validation | Base simulation | Observation | Cross-validation | Base simulation |
| China | **-6.2 (-12.0, -3.9)** [a] | **-4.9 (-10.3, -3.0)** | **-1.7 (-6.2, -0.8)** | -2.6 (-5.9, 0.1) | -2.1 (-5.9, 0.1) | **-0.9 (-3.0, -0.3)** |
| NCP | **-9.5 (-16.5, -7.2)** | **-8.1 (-14.5, -5.9)** | **-1.7 (-4.1, -1.4)** | -2.0 (-5.9, 0.0) | -2.1 (-5.6, 0.1) | **-0.6 (-1.6, -0.3)** |
| NE | **-6.8 (-14.6, -4.9)** | **-5.9 (-12.1, -4.1)** | **-1.8 (-7.6, -0.6)** | **-3.0 (-4.9, -1.1)** | **-3.3 (-5.4, -1.2)** | **-1.3 (-3.8, -0.3)** |
| SE | **-4.4 (-6.7, -2.5)** | **-3.7 (-5.6, -2.0)** | **-1.0 (-2.9, -0.1)** | -2.4 (-5.3, 0.1) | -2.5 (-5.1, 0.1) | **-1.0 (-1.8, -0.3)** |
| SW | **-4.2 (-8.8, -1.9)** | **-2.8 (-7.6, -1.3)** | **-3.4 (-15.6, -1.9)** | -1.8 (-6.2, 0.3) | -1.6 (-6.5, 0.2) | **-0.7 (-3.9, -0.2)** |
| NW | **-2.3 (-11.1, 0.6)** | **-4.2 (-7.7, -1.1)** | -1.9 (-13.7, 1.0) | -3.4 (-8.4, 2.3) | -1.7 (-9.5, 1.3) | -1.0 (-6.5, 0.3) |
| Central | **-7.9 (-17.5, -3.3)** | **-5.5 (-15.7, -2.3)** | -0.6 (-10.2, 0.0) | -2.0 (-6.6, 1.9) | -1.0 (-8.0, 2.2) | -0.5 (-3.8, 0.1) |

| | CO (mg/m³) | | | O₃ (μg/m³) | | |
|---|---|---|---|---|---|---|
| | Observation | Cross-validation | Base simulation | Observation | Cross-validation | Base simulation |
| China | **-0.12 (-0.17, -0.06)** | **-0.12 (-0.18, -0.07)** | **-0.02 (-0.05 -0.01)** | **3.5 (2.1, 5.0)** | **3.8 (2.1, 5.0)** | **2.0 (0.1, 5.9)** |
| NCP | **-0.18 (-0.25, -0.11)** | **-0.17 (-0.24, -0.11)** | **-0.03 (-0.05, -0.02)** | **5.3 (2.5, 8.7)** | **5.5 (2.4, 8.8)** | 1.4 (-0.5, 5.0) |
| NE | **-0.13 (-0.21, -0.05)** | **-0.13 (-0.20, -0.06)** | **-0.03 (-0.07, -0.01)** | **4.8 (1.5, 10.0)** | **4.6 (1.4, 9.5)** | 2.8 (-0.4, 8.0) |
| SE | **-0.06 (-0.09, -0.04)** | **-0.06 (-0.08, -0.04)** | **-0.01 (-0.02, -0.01)** | **2.3 (0.3, 3.4)** | **2.6 (0.8, 3.5)** | **1.7 (0.3, 3.0)** |
| SW | **-0.11 (-0.19, -0.04)** | **-0.09 (-0.21, -0.04)** | **-0.02 (-0.06, -0.01)** | **3.2 (1.2, 5.0)** | **3.5 (1.8, 5.4)** | 2.7 (-0.9, 7.1) |
| NW | -0.14 (-0.46, 0.04) | -0.14 (-0.30, 0.04) | -0.03 (-0.06, 0.00) | **5.4 (1.6, 9.8)** | **4.0 (1.4, 10.1)** | 2.6 (-0.2, 8.8) |
| Central | **-0.16 (-0.27, -0.09)** | **-0.17 (-0.25, -0.10)** | -0.01 (-0.06, 0.00) | **5.3 (2.3, 9.2)** | **4.5 (1.4, 7.8)** | 2.2 (-0.3, 7.7) |

[a] The bold font denotes that the calculated trend is significant at the 0.05 significance level, and the values in brackets denote the 95% confidence interval.

**Table 7: Comparison of the data accuracy of CAQRA and CAMSRA in China**

| | CAQRA | | | | CAMSRA | | | |
|---|---|---|---|---|---|---|---|---|
| | $SO_2$ ($\mu g/m^3$) | $NO_2$ ($\mu g/m^3$) | CO ($mg/m^3$) | $O_3$ ($\mu g/m^3$) | $SO_2$ ($\mu g/m^3$) | $NO_2$ ($\mu g/m^3$) | CO ($mg/m^3$) | $O_3$ ($\mu g/m^3$) |
| $R^2$ | 0.53 | 0.61 | 0.55 | 0.77 | 0.04 | 0.23 | 0.13 | 0.00 |
| MBE | -2.0 | -2.3 | -0.1 | -2.3 | 19.4 | 1.7 | -0.2 | 30.6 |
| NMB (%) | -8.5 | -6.9 | -6.1 | -4.0 | 81.2 | 5.2 | -17.5 | 52.1 |
| RMSE | 24.8 | 16.4 | 0.5 | 21.9 | 54.5 | 27.3 | 0.9 | 55.2 |

























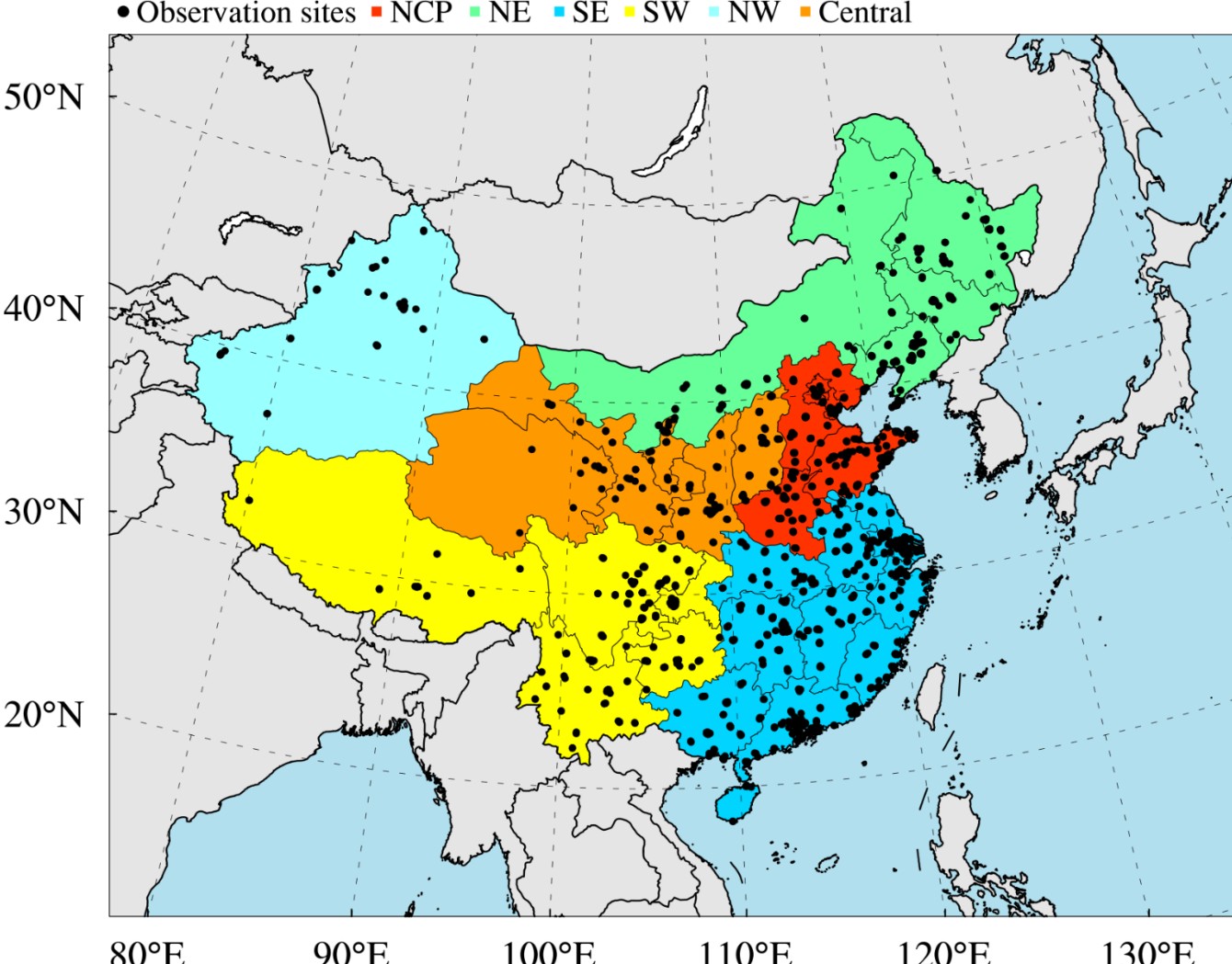

 **Figure 1: Modelling domain of the ensemble simulation overlain on the distribution of the observation sites of the CNEMC.   The**
**different colours denote the different regions in China, namely, the North China Plain (NCP), Northeast China (NE), Southwest**
**China (SW), Southeast China (SE), Northwest China (NW) and Central China.**


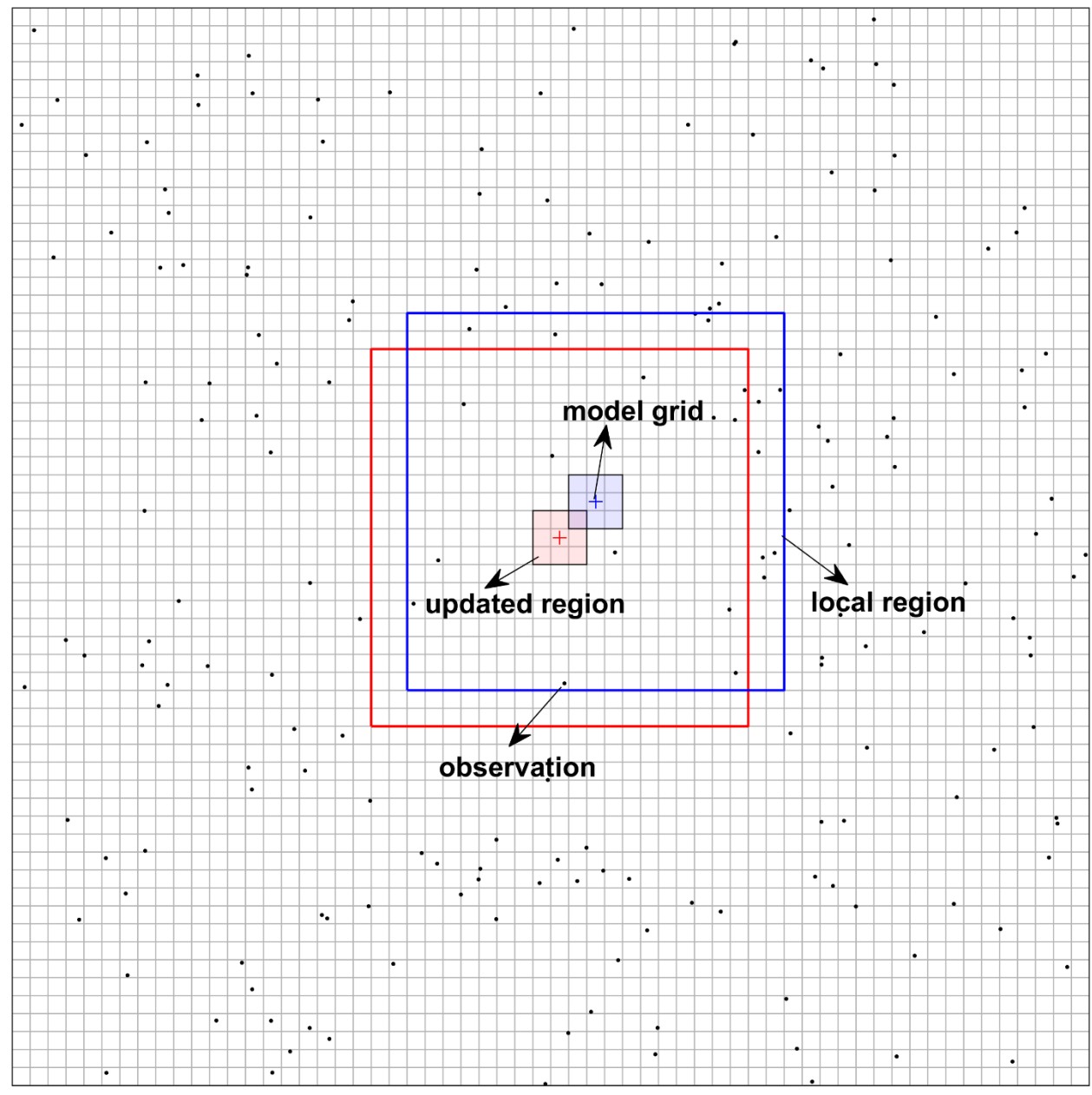

**Figure 2: Illustration of the local analysis scheme used in the assimilation. The plus and dot symbols denote the centres of the model grids and the location of the observation sites, respectively. The large rectangular region denotes the local region, and the shaded region denotes the updated region.**




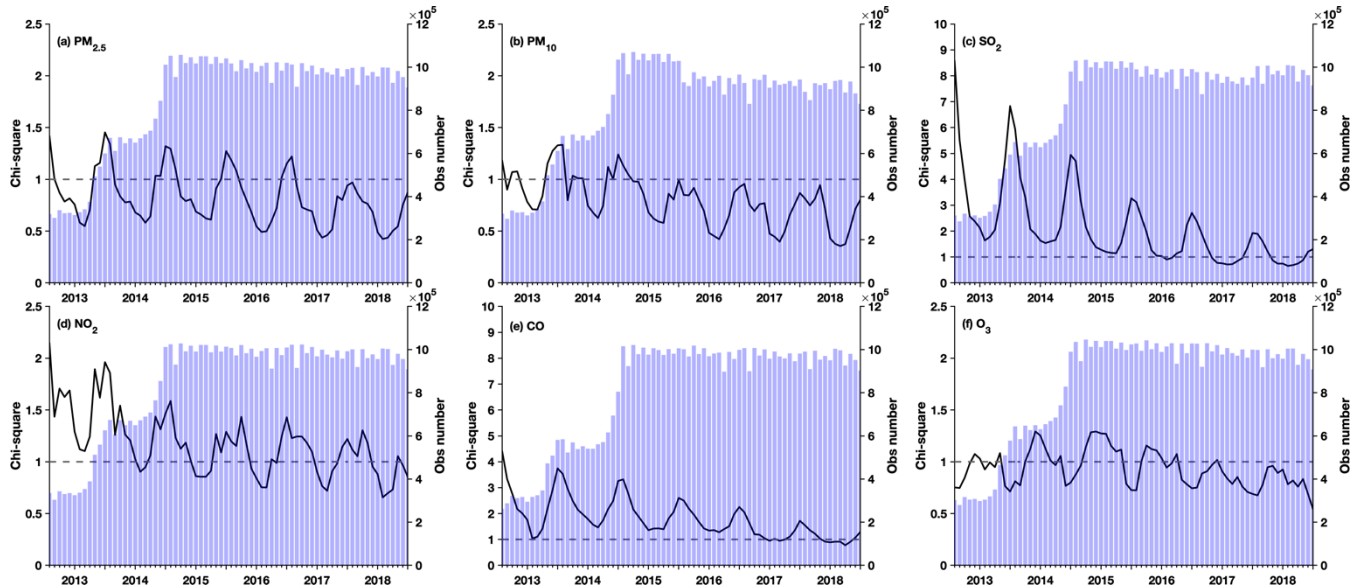

**Figure 3: Time series of the monthly mean $\chi^2$ values (black line) and the number of assimilated observations per month (blue bars) for (a) PM$_{2.5}$, (b) PM$_{10}$, (c) SO$_2$, (d) NO$_2$, (e) CO and (f) O$_3$.**

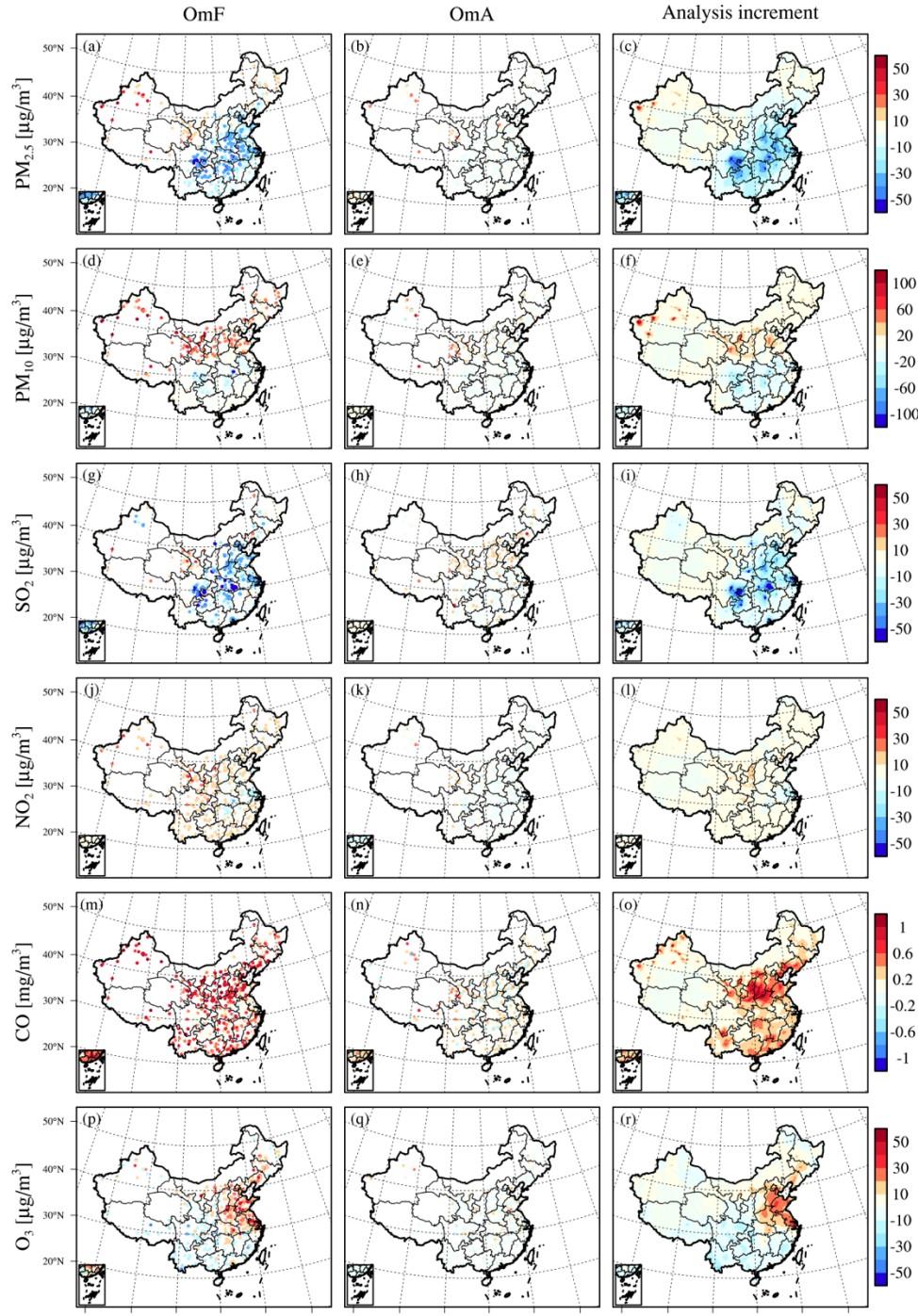


**Figure 4: Spatial distributions of the six-year mean OmF (left panel), OmA (middle panel) and analysis increment (right panel) for different species in China.**

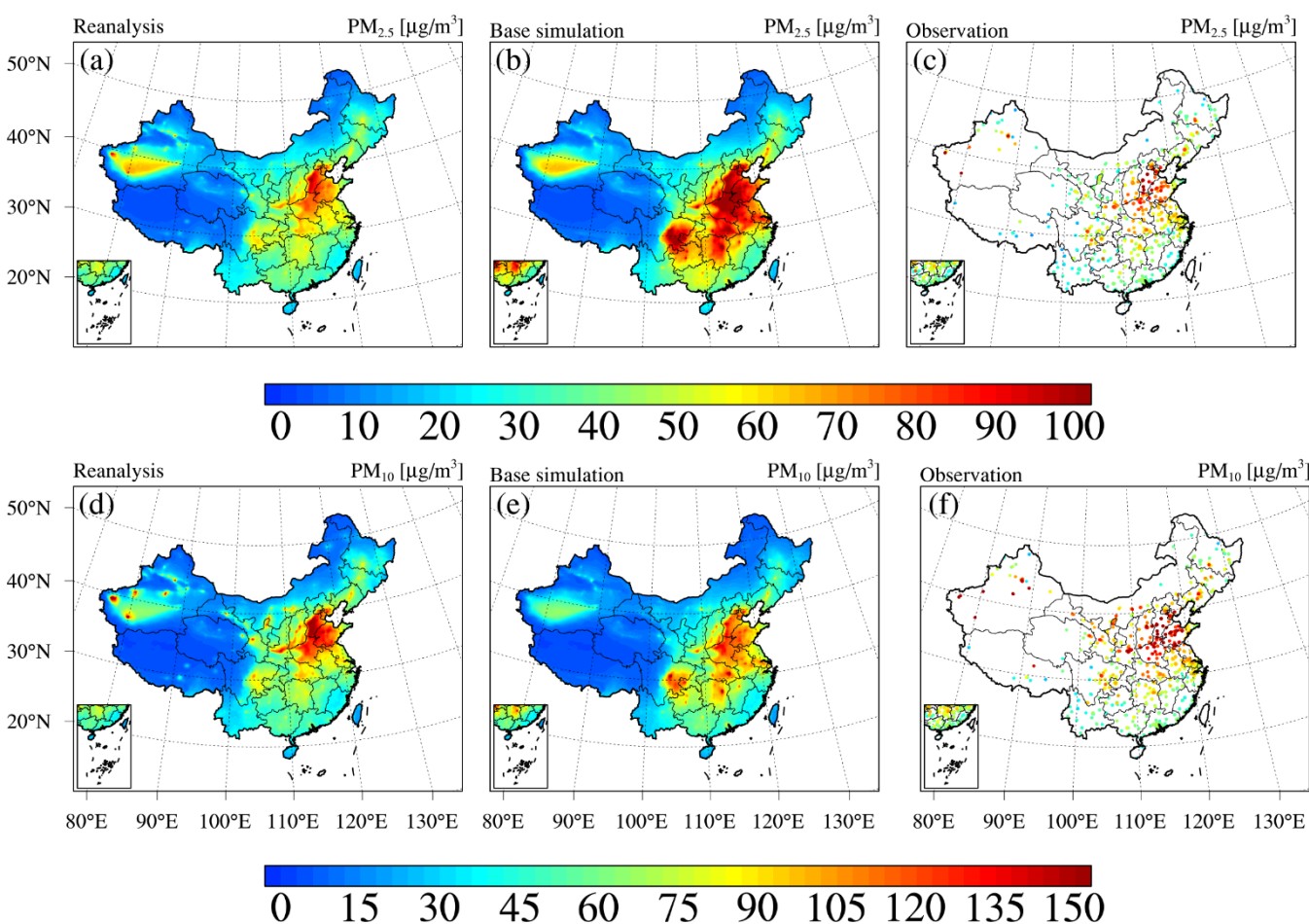

**Figure 5: Spatial distributions of the (a–c) PM₂.₅ and (d–f) PM₁₀ concentrations in China from (a, d) CAQRA, (b, e)**
**base simulation and (c, f) observations averaged from 2013 to 2018.**

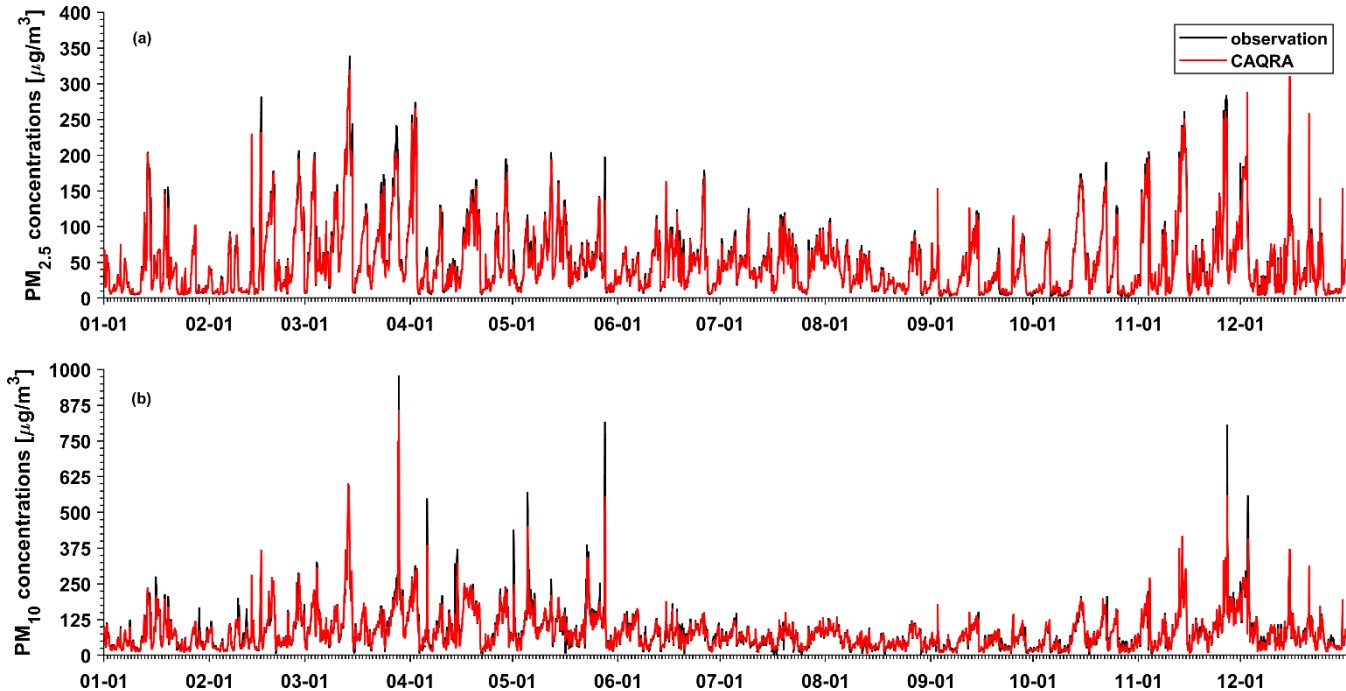

Figure 6: Time series of the site mean hourly (a) PM$_{2.5}$ and (b) PM$_{10}$ concentrations in Beijing obtained from the observations and CAQRA.

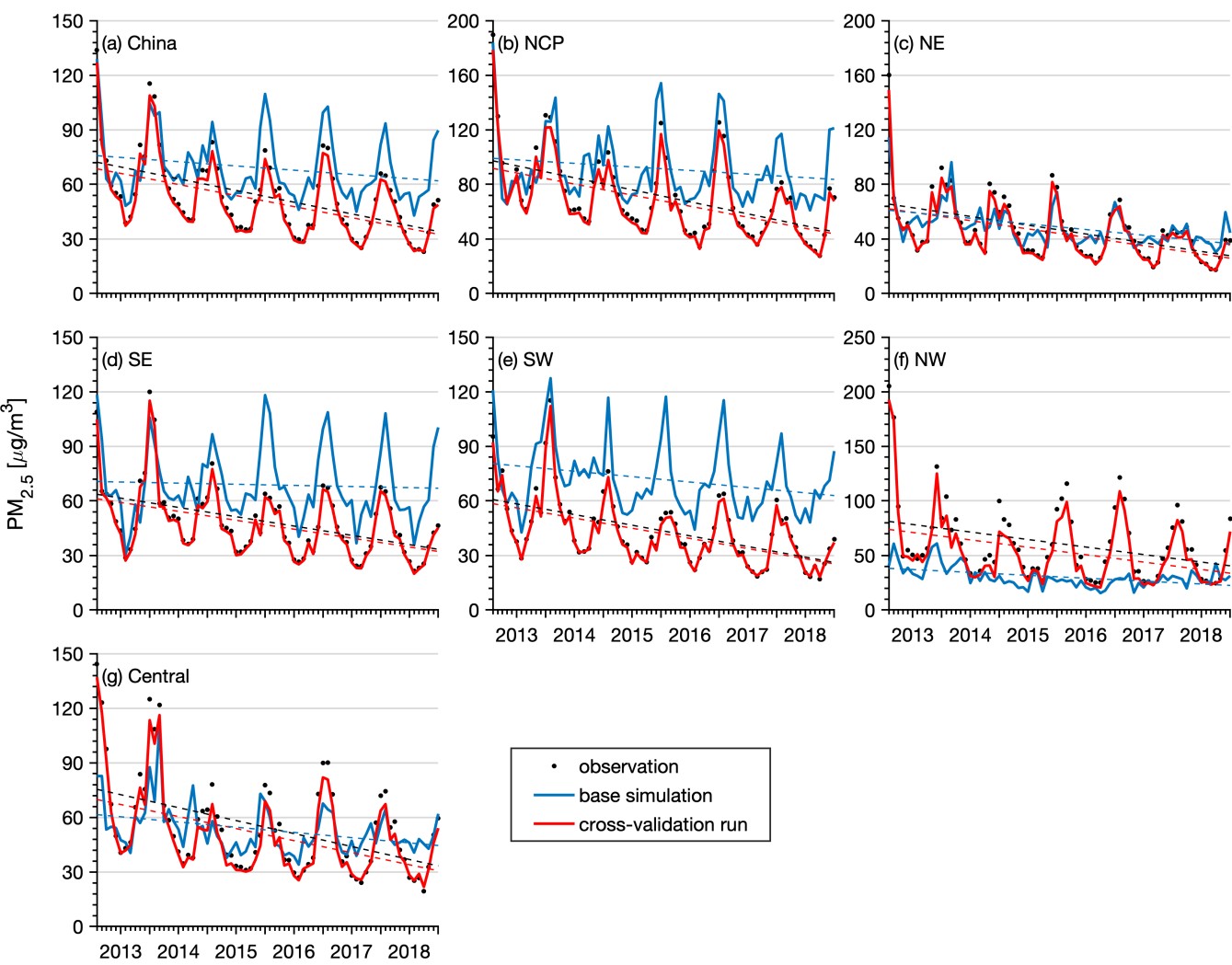

**Figure 7: Time series of the monthly mean PM₂.₅ concentrations in (a) China, (b) NCP, (c) NE, (d) SE, (e) SW, (f) NW and (f) central regions obtained from the cross-validation run (red line), base simulation (blue line) and observations (black dots).**

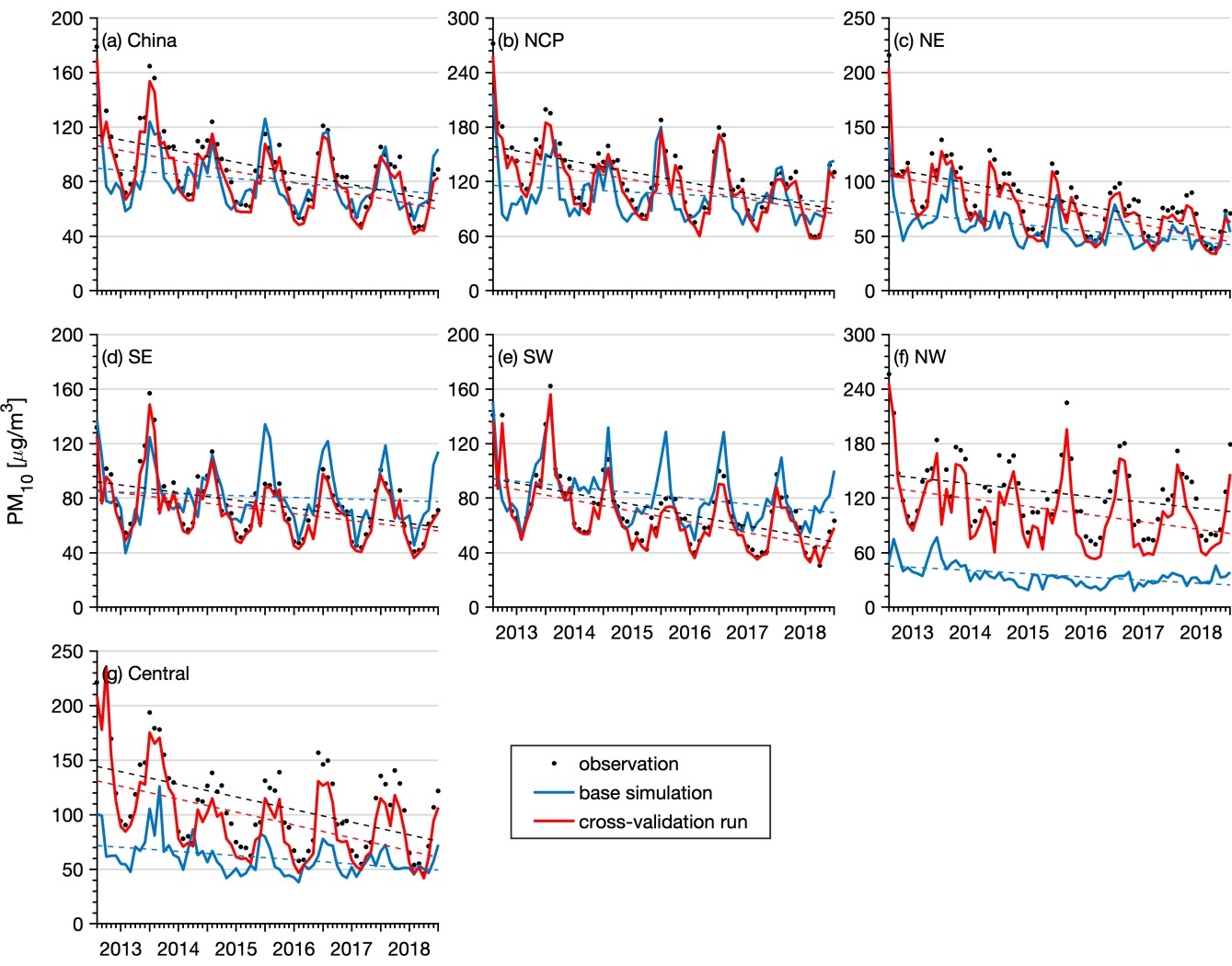


**Figure 8: Same as Fig. 7 but for the PM₁₀ concentration.**

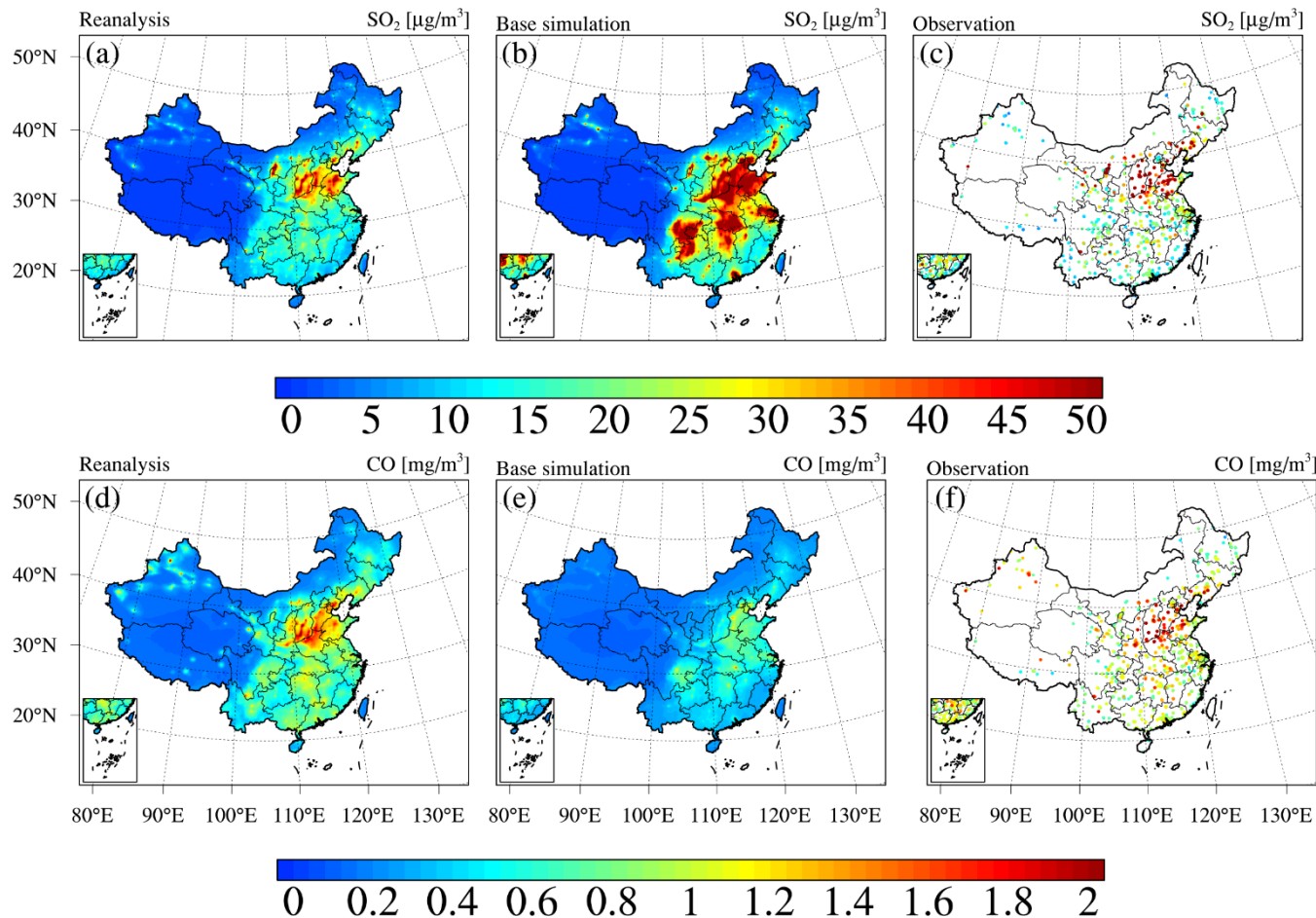


**Figure 9: Same as Fig. 5 but for the SO₂ and CO concentrations.**

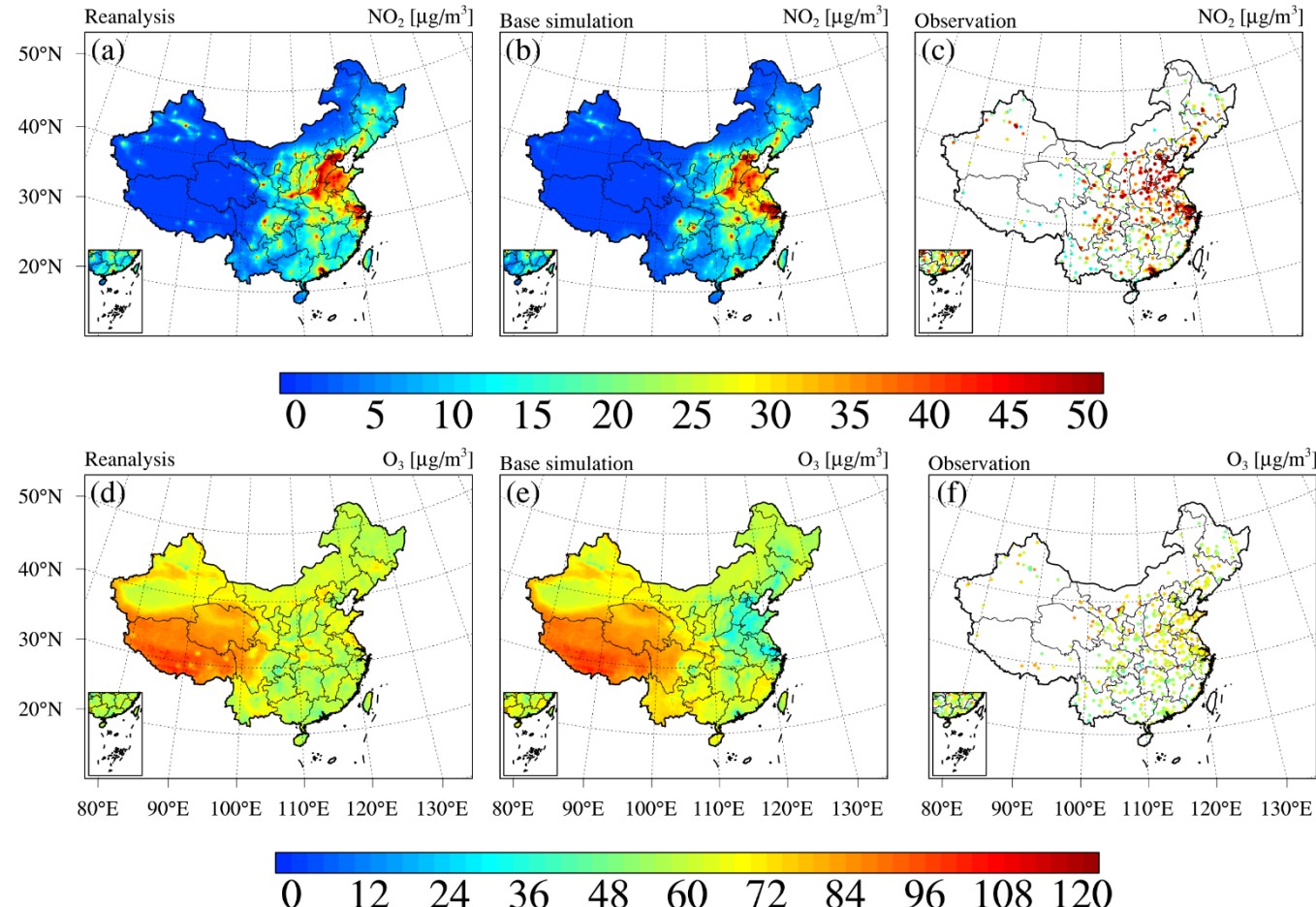

Figure 10: Same as Fig. 5 but for NO₂ and O₃.

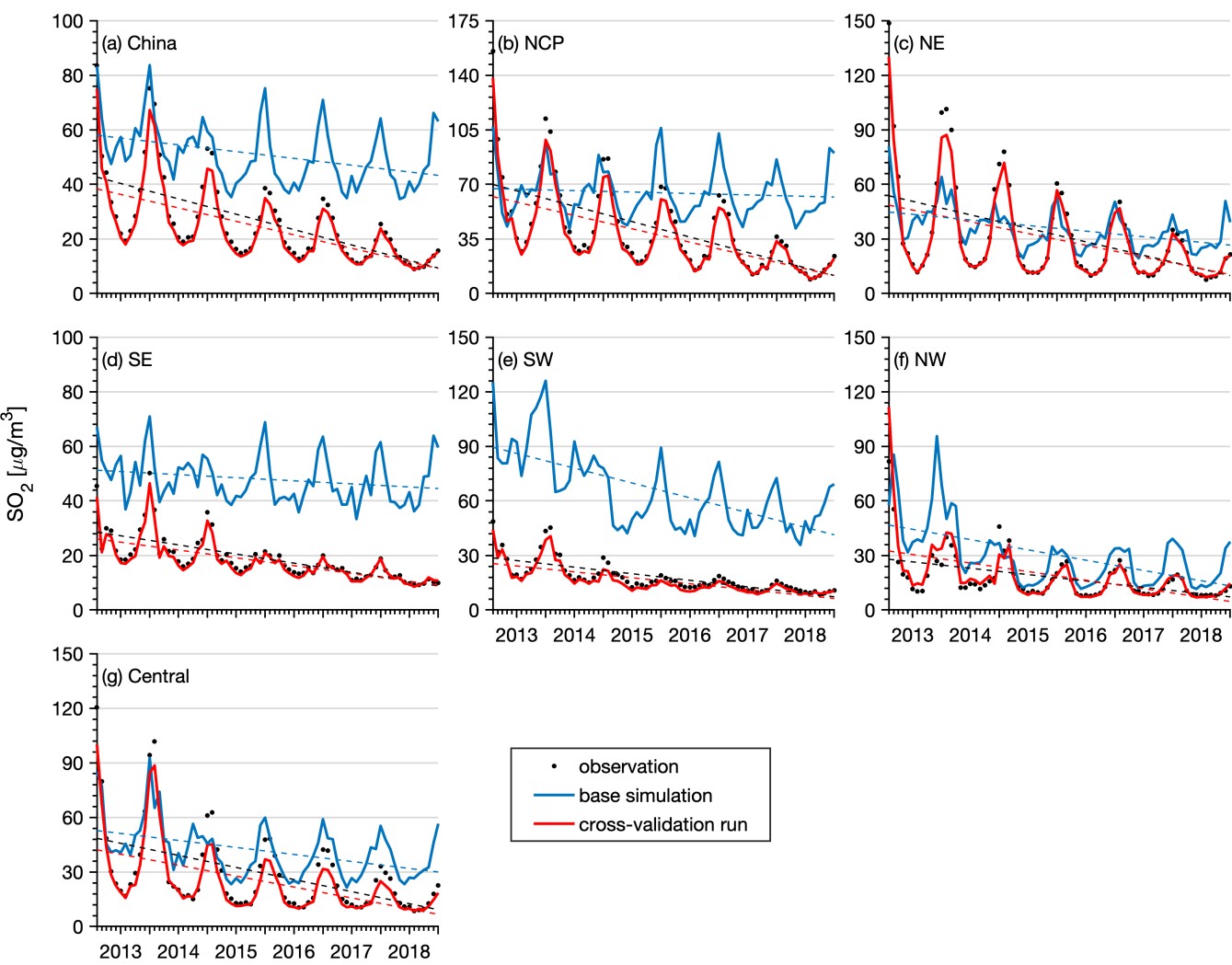


**Figure 11: Same as Fig. 7 but for the SO₂ concentration.**

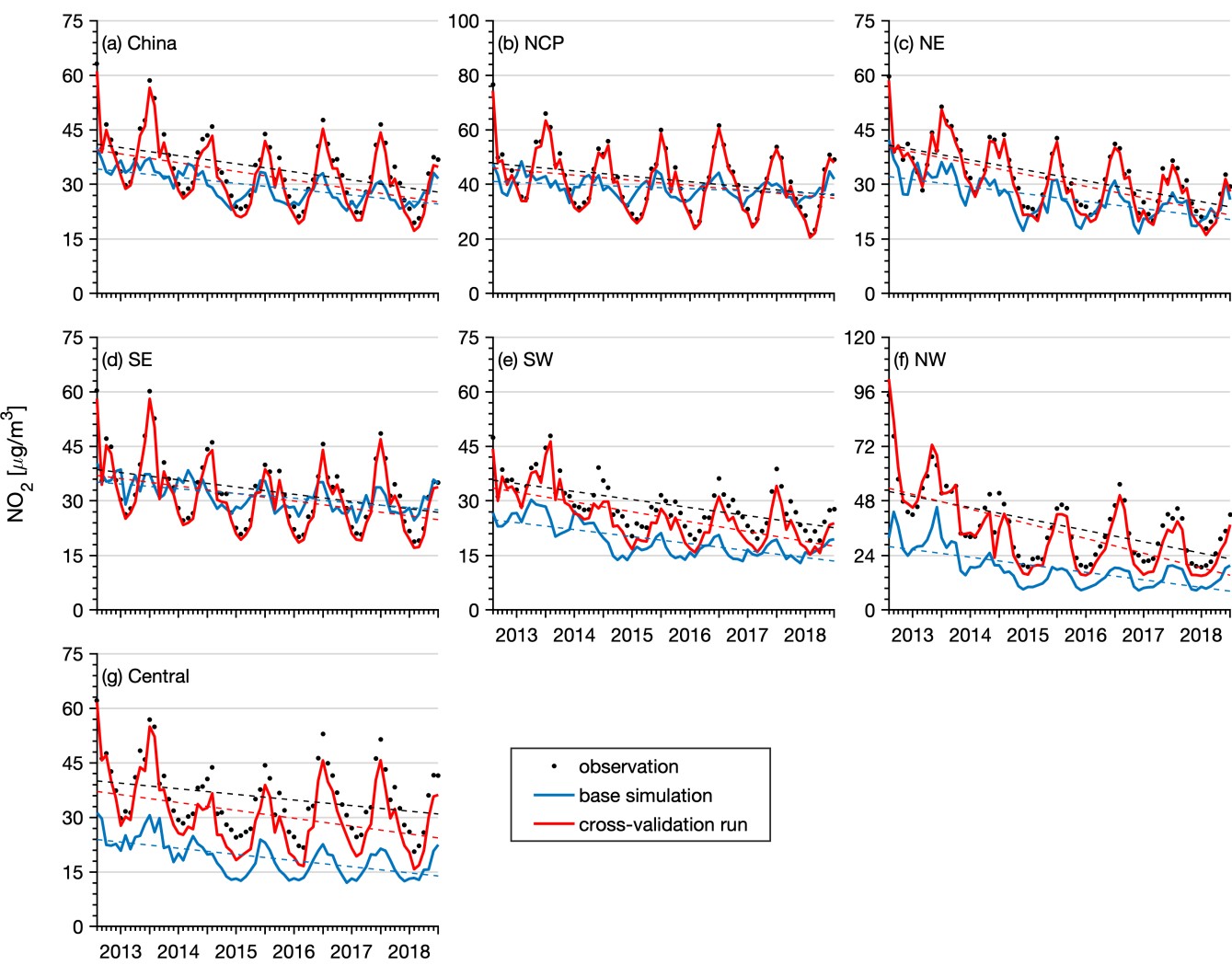


**Figure 12: Same as Fig. 7 but for the NO₂ concentration.**

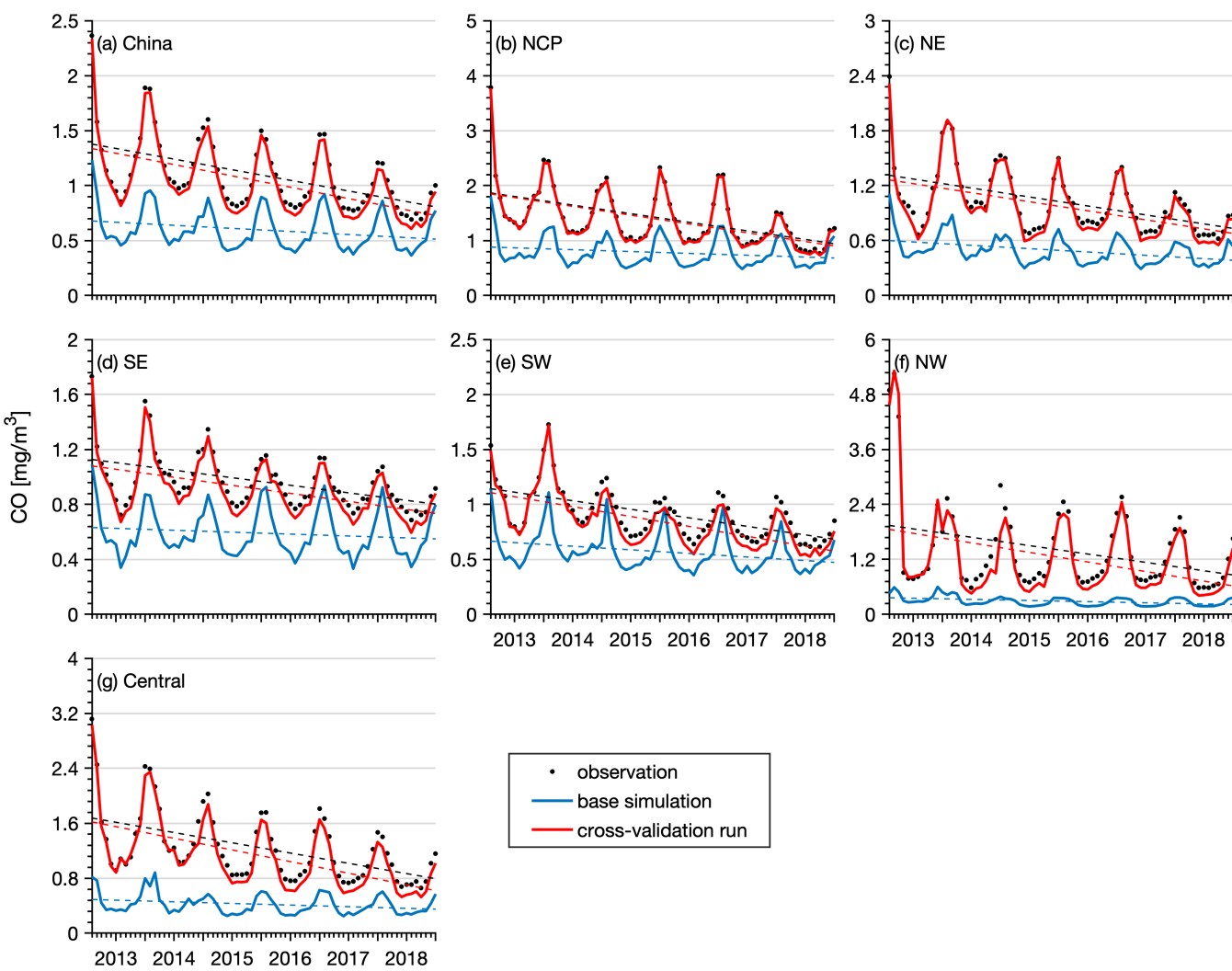


**Figure 13: Same as Fig. 7 but for the CO concentration.**

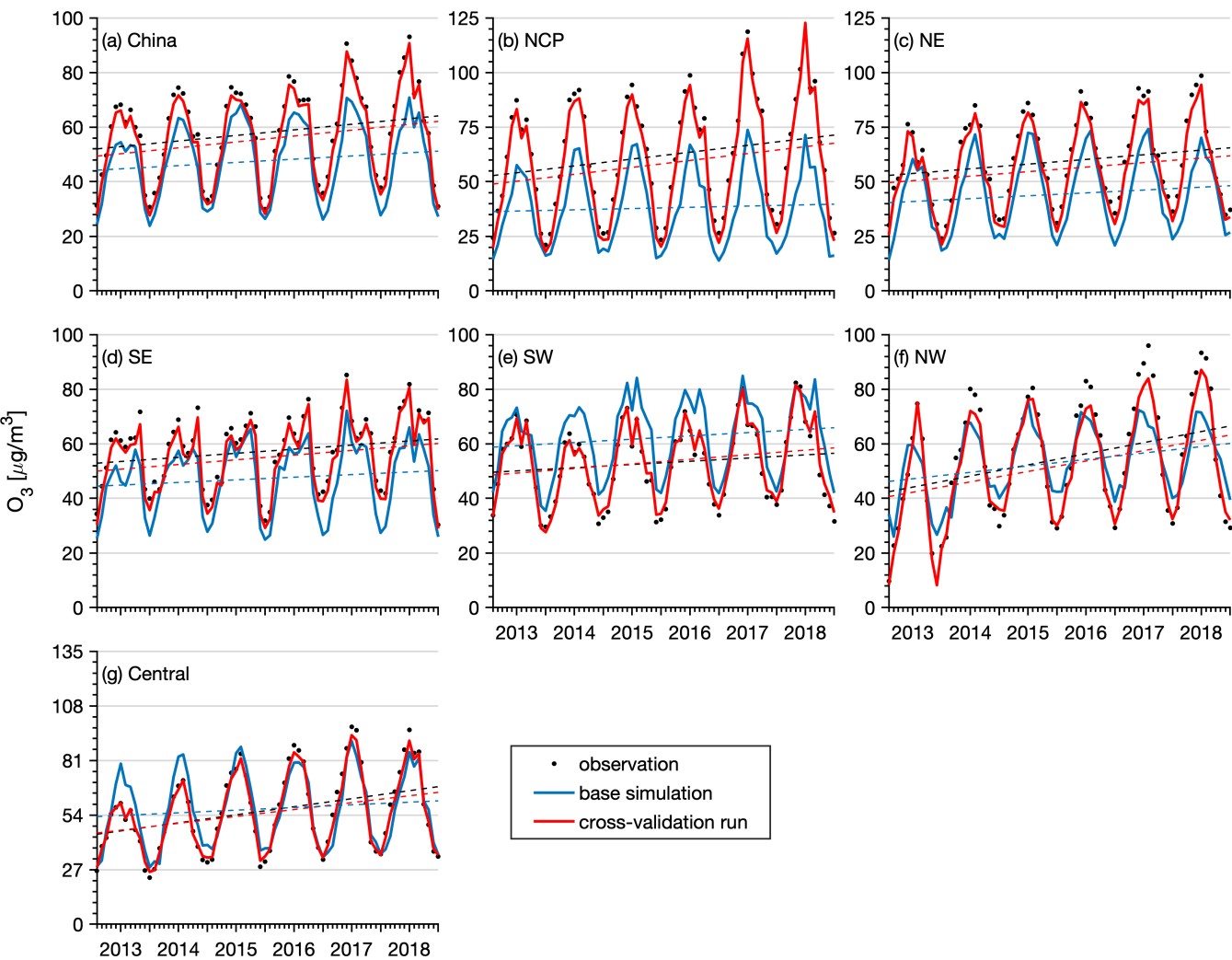


**Figure 14: Same as Fig. 7 but for the O₃ concentration.**

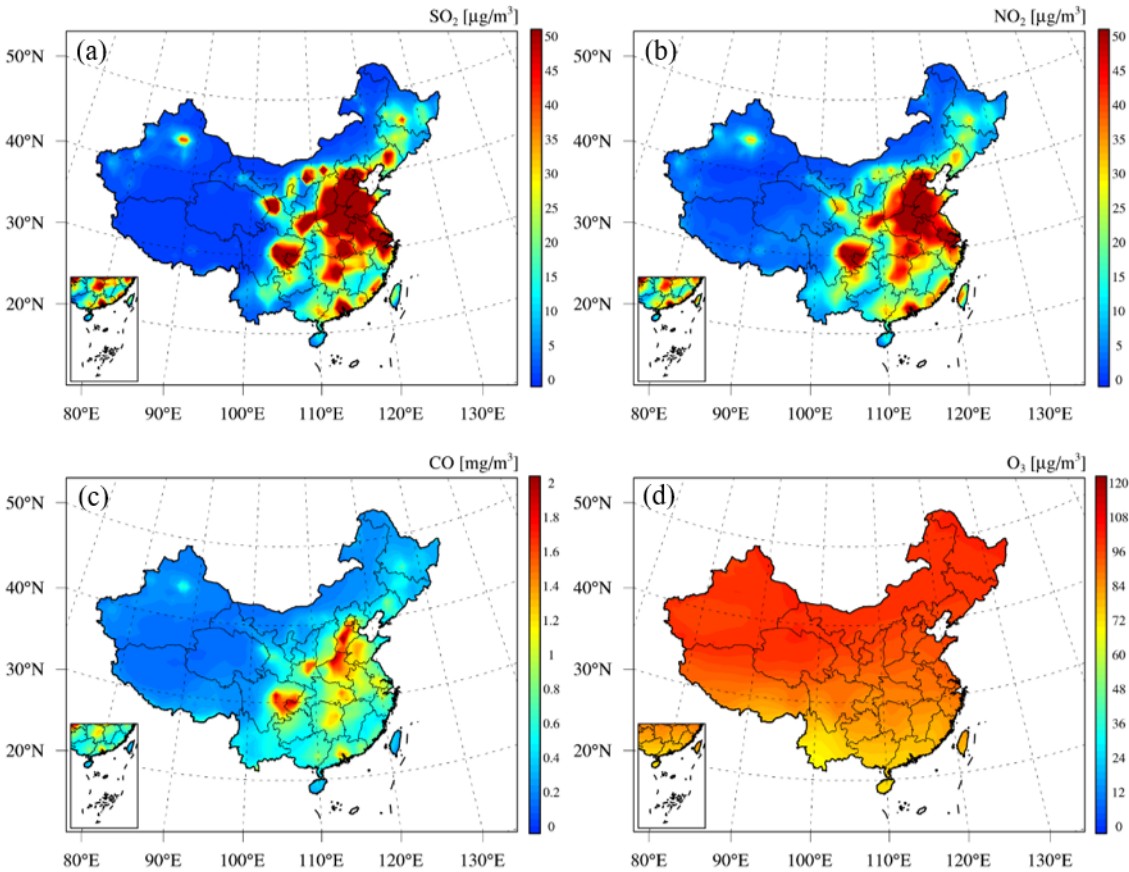


**Figure 15: Spatial distributions of the multiyear average concentrations of (a) SO₂, (b) NO₂, (c) CO and (d) O₃ from**
**2013 to 2018 obtained from CAMSRA.**

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
