# Peer review of "A six-year-long (2013–2018) high-resolution air quality reanalysis"

_Earth System Science Data, 2020_

## Referee Comment (RC1) · Anonymous Referee #1 · 24 Jul 2020

This study presents high-resolution air quality reanalysis products over China for 2013-2018. The air quality reanalysis assimilated the country-wide surface observations using the regional EnKF data assimilation. The assimilated results were evaluated against the assimilated and independent measurements. The topic of this study is very interesting, and the produced data sets can be useful for various applications. The paper is generally well written. However, because this is the first paper describing the system and data, more careful description of the system and its performance would be useful for readers and future developments. Please see my recommendations below.

[Figure]

1. The representativeness error estimation is not clear. How did you estimate L_repr for each station and Epsilon^abs for each species? Urban and rural observations could be (or should be) used in a different way, but this is not mentioned. Were any temporal averages applied to the observations? Temporal variability information could be used a part of representativeness errors. Further explanation is needed.

2. The assimilated results are compared with the independent observations for PM but with the assimilated observations only for other species (they only demonstrate self-consistency. CAMS is not observation). This provides limited information on the performance of the developed system. The Chi-square diagnostic can be used to see whether the Kalman filtering worked properly. OmF & OmA statistics can also be demonstrated. Given limited validation data, more efforts are required to demonstrate the performance.

3. Inter-species correlation was totally neglected in background error covariance. This setting is extremely conservative and does not fully utilize the advantages of EnKF data assimilation that produces comprehensive background error patterns. I'm wondering if the authors have tried to implement inter-species correlations. Further discussion is needed (e.g., why it is so conservative, what is the disadvantage of the current setting).

4. Please clarify whether there are any variations in inflation factor and how it was optimized for different species. In most regional ensemble data assimilation systems, fixed lateral boundary condition tends to limit the effectiveness of data assimilation near their boundaries (and also inside when horizontal advection is strong) because of reduced spreads. Did you find any problem with it?

5. Using automatic outlier detection method, how much observations were rejected? What was the impact in data assimilation?

6. Because of the fine-scale variability and large degree of freedoms, the high-region data assimilation would require larger ensembles. I'm wondering if 50 members are sufficient. Further discussion is needed to demonstrate whether the background error

is produced properly to propagate observational information in space.

---

## Referee Comment (RC2) · Anonymous Referee #2 · 29 Nov 2020

This paper presents a six-year reanalysis of air quality in China, providing data for six criteria air pollutants at 15 km spatial resolution and hourly temporal resolution. This is a unique data set and the methods for producing it appear to be sound. I presume that there will be some demand for the data set, although it would have been good to get a more specific identification of stakeholders from the paper.

The main problem I have with the paper is that the writing is atrocious. The title is ungrammatical (should 'base" be "based"?), the first sentence of the abstract is vapid, and it doesn't improve from there. The paper is basically unreadable. I don't know to

what extent this matters for an ESSD publication, but as a reader I would feel that if the paper is that bad then the dataset must be bad too. It's not clear to me what to do about this except to request that the authors rewrite their paper to abide by grammatical, clear, and concise language - I will leave it to the Editor to decide whether this is an appropriate request.

---

## Author Comment (AC1) · 27 Dec 2020

We Thank Reviewer for his/her constructive comments.

Responses to the Specific comments:

**General comments:** This study presents high-resolution air quality reanalysis products over China for 2013-2018. The air quality reanalysis assimilated the country-wide surface observations using the regional EnKF data assimilation. The assimilated results were evaluated against the assimilated and independent measurements. The topic of this study is very interesting, and the produced data sets can be useful for various applications. The paper is generally well written. However, because this is the first paper describing the system and data, more careful description of the system and its performance would be useful for readers and future developments.

**Reply:** The authors appreciate the reviewer for his/her constructive and up-to-point comments. We have carefully considered the comments and revised the manuscript accordingly. Please refer to our responses for more details given below.

**Comment 1:** The representativeness error estimation is not clear. How did you estimate L\_repr for each station and  $\varepsilon^{abs}$  for each species? Urban and rural observations could be (or should be) used in a different way, but this is not mentioned. Were any temporal averages applied to the observations? Temporal variability information could be used a part of representativeness errors. Further explanation is needed.

**Reply:** Thanks for this important suggestion. The representativeness error arises from the different spatial scales that the gridded model results and discrete observations represent, which is parameterized by the formula proposed by Elbern et al. (2007) in this study:

$$r_{repr} = \sqrt{\frac{\Delta x}{L_{repr}}} \times \epsilon^{abs} \tag{1}$$

where  $r_{repr}$  represents the representativeness error,  $\Delta x$  represents the model resolution,  $L_{repr}$  represents the characteristic representativeness length of the observation site and  $\varepsilon^{abs}$  represents the error characteristic parameters for different species.

We agree with the reviewer that the  $L_{repr}$  should be treated differently for urban and rural sites since the urban sites usually have smaller representativeness length than the rural sites due to the larger representativeness error. According to Elbern et al. (2007), the representativeness length of urban and rural sites were 2km and 10km. Considering that the observation sites from CNEMC were almost city (urban) sites (>90%), the  $L_{repr}$  was assigned to be 2km in this study for simplicity.

For the estimations of  $\varepsilon^{abs}$ , previous studies (Chen et al., 2019; Feng et al., 2018; Jiang et al., 2013; Ma et al., 2019; Pagowski and Grell, 2012; Peng et al., 2017; Werner et al., 2019) usually assigned the  $\varepsilon^{abs}$  empirically

to be half of the measurement error following the study by Pagowski et al. (2010). In this study, the  $\varepsilon^{abs}$  was obtained from Li et al. (2019) who estimated the  $\varepsilon^{abs}$  based on a dense observation network in Beijing-Tianjin-Hebei region. In their study, the representativeness error of each species' observation was first estimated by the spatiotemporal averaged standard deviation of the observed values within a 30km×30km grid:

$$r_{repr,i} = \frac{1}{MT} \sum_{m=1}^{M} \sum_{t=1}^{T} S_{m,t,i}$$
(2)

where  $r_{repr,i}$  represents the representativeness errors of the observations for species *i*,  $S_{m,t,i}$  represents the standard deviation of the observed values of species *i* at different sites that are located in a same grid *m* at time *t*, *M* and *T* represents the total number of grid and observation time. After that, the  $\varepsilon_i^{abs}$  for species *i* were estimated by a transformation of Eq. (1):

$$\varepsilon_i^{abs} = r_{repr,i} / \sqrt{\frac{\Delta x}{L_{repr}}}$$
(3)

where  $\Delta x$  is equal to 30km. Based on the estimated  $L_{repr}$  and the  $\varepsilon_i^{abs}$  for different species, the representativeness errors are estimated using Eq. (1) by specifying the  $\Delta x$  to be 15km. Following the suggestions of the reviewer, we have added more explanation to the estimations of representativeness error in the revised manuscript (*please see lines 223–245 in the revised manuscript*).

**Comment 2:** The assimilated results are compared with the independent observations for PM but with the assimilated observations only for other species (they only demonstrate self-consistency. CAMS is not observation). This provides limited information on the performance of the developed system. The Chi-square diagnostic can be used to see whether the Kalman filtering worked properly. OmF & OmA statistics can also be demonstrated. Given limited validation data, more efforts are required to demonstrate the performance.

**Reply:** Thanks for this important comment. Following the suggestions of reviewer, we have added the analysis of  $\chi^2$  diagnosis and the statistics of observation minus forecast (OmF:  $y^o - H(x^b)$ ) & observation minus analysis (OmA:  $y^o - H(x^a)$  in the revised manuscript to demonstrate the performance of our assimilation system (*please see lines 317–369 in the revised manuscript*).

 $\chi^2$  diagnosis is a robust criterion for validating the estimated background and observation error covariance in the data assimilation (e.g., Menard et al., 2000; Miyazaki et al., 2015; Miyazaki et al., 2012), which is estimated by comparing the sample covariance of OmF with the sum of estimated background and observation error covariance in the observational space (**HBH**T + **R**):

$$Y = \frac{1}{\sqrt{m}} (HBH^{T} + R)^{-\frac{1}{2}} (y^{o} - HX^{b})$$

$$\chi^{2} = Y^{T}Y$$
(5)

where *m* is the number of observations. According to the Kalman filtering theory, the mean of  $\chi^2$  should approach

1 if the background and observation error covariances are properly specified, while values greater (lower) than 1 indicates the underestimation (overestimation) of the observation and/or background error covariance.

Figure R1 shows the time series of the monthly  $\chi^2$  values (black lines) for different species as well as the number of assimilated observations per month (blue bars). The mean values of  $\chi^2$  are generally within 50% difference from the ideal value of 1 for PM2.5, PM10, NO2 and O3, which suggests that the observation and background error covariance are generally well specified in the analysis of these species. Although the  $\chi^2$  values for these species showed pronounced seasonal variations that reflects the different error characteristics in different seasons, the  $\chi^2$  values were roughly stable for PM2.5 and O3 throughout the period, and for NO2 and PM10 after 2015 when the number of assimilated observations become stable, which generally shows the long-term stability of the performance of data assimilation. The  $\chi^2$  values for SO2 were nevertheless greater than 1 in most cases, especially before 2017. This would be more relevant to the underestimations of background error covariance of SO2 as we only specified 12% uncertainty in the SO2 emissions. suggesting that the emission uncertainty of SO2 may be underestimated by Zhang et al. (2009). There were also pronounced annual trends in the  $\chi^2$  values of SO2, which may be attributed to the increases of observation number from 2013 to 2014 and the substantial decreases of SO2 observations. Although smaller than the  $\chi^2$  values of SO2, the values for CO were greater than 1 in most cases, suggesting the underestimations of the error covariances. Obvious decreasing trend can also be found in the  $\chi^2$  values of CO. The  $\chi^2$  test results suggest that our data assimilation system has relatively poor performance in the analysis of CO and SO2 concentrations than the other four species, which is consistent with the cross-validation results which showed smaller  $R^2$  values for the reanalysis data of CO and SO2 concentrations (Sect. 4.2.2 in the *revised manuscript*). The annual trend of  $\chi^2$  values in CO and SO2 also indicates relatively weak stability in the performance of data assimilation system on assimilating CO and SO2 observations, which may influence the analysis of the annual trends in these two species. Based on these results, we have added discussions on this issue in our revised manuscript to inform the potential users of the problems that they should be aware of (please see lines 667 – 670 in the revised manuscript).

---

## Author Comment (AC2) · 27 Dec 2020

**Response to Referee #2 (essd-2020-100)**

We Thank Reviewer for his/her constructive comments.

Responses to the Specific comments:

**General comments:** This paper presents a six-year reanalysis of air quality in China, providing data for six criteria air pollutants at 15 km spatial resolution and hourly temporal resolution. This is a unique data set and the methods for producing it appear to be sound. I presume that there will be some demand for the data set, although it would have been good to get a more specific identification of stakeholders from the paper. The main problem I have with the paper is that the writing is atrocious. The title is ungrammatical (should 'base" be "based"?), the first sentence of the abstract is vapid, and it doesn't improve from there. The paper is basically unreadable. I don't know to what extent this matters for an ESSD publication, but as a reader I would feel that if the paper is that bad then the dataset must be bad too. It's not clear to me what to do about this except to request that the authors rewrite their paper to abide by grammatical, clear, and concise language - I will leave it to the Editor to decide whether this is an appropriate request.

Reply: We sincerely apologize for the poor presentation quality. A thorough revision has been made to improve the language of the paper. The paper was edited for grammar, phrasing, and punctuation. In addition, many edits were made to further improve the flow and readability of the text. Specifically, A variety of edits were made to ensure smooth transitions between sentences and to link related thoughts. Sentence flow is improved in the revised manuscript by ensuring the appropriate use of conjunctions and introductory words and phrases. Certain edits were made to remove redundant, repetitive or unnecessary phrasing and to present the information in a more straightforward manner. Some edits were also made to improve conciseness by trimming unnecessary words and streamlining the flow of the manuscript. The manuscript was also been polished by highly qualified native English speaking editors at American Chemical Society (Fig. R1). After these revisions, we believe the language of the revised manuscript can meet the requirement of the publication in ESSD.

Following the suggestion of reviewer, the potential usages of our reanalysis data are also emphasized in the revised manuscript to get a more specific identification of stakeholders from the paper. For example, the dataset can be used in the retrospective air quality analysis in China, health and environmental impact assessment of air pollution at fine scales, model evaluation and satellite calibration, optimization of monitoring sites and provision of basic training datasets for statistical or artificial intelligence (AI)-based forecasting (*please see lines 41–45 and lines 113–116*).

[Figure]

**Figure R1: The editing certificate of the revise manuscript provided by ACS.**

---

## Author Response (AR2)

**Response to Topical Editor (essd-2020-100)**

We thank the topical editor for his decision and suggestions.

Responses to the specific comments:

**Comment 1:** Thank you for making substantive changes in response to reviewer comments. I suggest that authors relabel the sections currently called 'Supplementary' as 'Appendices'; they definitely want to keep this set of information with the main manuscript. Please also inform users about file sizes and file structures in the data availability section. The Science DB links work very well (thank you) but please give users some guidance on what they will find and what they might download for what purposes.

**Reply:** Thanks very much for your decision and suggestions. We have relabelled the 'Supplementary' sections as the 'Appendices' in the revised manuscript (*please see lines 699–827 in the revised manuscript*). Following the suggestions of editor, we also rewritten the data availability section to inform users more information about the reanalysis dataset, including the total file sizes, file structure, and some guidance on the use of Science DB links (*please see lines 620–631 in the revised manuscript*).

**Changes in manuscript:** lines 203, 209, 281, 356, 361, 403, 428–429, 528–529, 546, 620–631, 699–827 and 845–847.

[revised manuscript text omitted]

descriptions of CAQRA dataset, and 2192 zip files listed in the DATA FILES column at the website. The total file sizes are
approximately 318.81 GB as of the time of this writing. Each zip file is named by the date and contains one day's reanalysis
data, which is composed of 24 Network Common Data Form (NetCDF) files. Each NetCDF file contains one hour's reanalysis
data and is named by the date. The time zone of the reanalysis data is Beijing Time and the description on the content of each
NetCDF file is available in README.txt at the website. The monthly and annual version of the CAQRA dataset contains two
zip files which are the monthly and annual mean of the reanalysis data, respectively. The total file sizes of this product are
approximately 480.67MB which is more easily to be downloaded and suitable for users who only need air quality data at
monthly or yearly scales.

[revised manuscript text omitted]

**quality control averaged from 2013 to 2018.**

 **Appendix B: Inter-species correlation coefficient among different species**

**inter-species correlation coefficient**

| | $PM_{2.5}$ | $PM_{10}$ | $SO_2$ | $NO_2$ | CO | $O_3$ |
|---|---|---|---|---|---|---|
| $PM_{2.5}$ | 1.00 | 0.94 | 0.04 | -0.01 | 0.02 | -0.02 |
| $PM_{10}$ | 0.94 | 1.00 | 0.04 | -0.01 | 0.02 | -0.02 |
| $SO_2$ | 0.04 | 0.04 | 1.00 | 0.05 | -0.01 | -0.06 |
| $NO_2$ | -0.01 | -0.01 | 0.05 | 1.00 | 0.03 | -0.63 |
| CO | 0.02 | 0.02 | -0.01 | 0.03 | 1.00 | 0.07 |
| $O_3$ | -0.02 | -0.02 | -0.06 | -0.63 | 0.07 | 1.00 |

**Figure B1: Correlations between species in the background error covariance matrix, estimated from the LETKF**

**ensemble averaged from 2013 to 2018. The global mean of the covariance estimated for each station is plotted.**

[Figure]

**Figure B2: Correlations between species in the background error covariance matrix, estimated from the LETKF ensemble averaged in different seasons from 2013 to 2018. The global mean of the covariance estimated for each station is plotted.**

 **Appendix C: Time series of the OmF and OmA statistics from the data assimilation system**

[Figure]

**Figure C1: Time series of monthly mean OmF and OmA normalized mean bias in different regions of China for**
**different species.**

[Figure]

**Figure C2: Time series of monthly mean OmF and OmA normalized root mean square error in different regions of**

**China for different species.**

 **Appendix D: Spatial distributions of seasonal mean concentrations of different species obtained from CAQRA**

[Figure]

**Figure D1: Spatial distributions of the PM2.5 reanalysis in China during (a) spring, (b) summer, (c) autumn and (d) winter averaged from 2013 to 2018.**

[Figure]

**Figure D2: Same as Fig. D1 but for PM₁₀.**

[Figure]

**Figure D3: Same as Fig. D1 but for SO₂.**

[Figure]

**Figure D4: Same as Fig. D1 but for CO.**

[Figure]

Figure D5: Same as Fig. D1 but for NO₂.

[Figure]

Figure D6: Same as Fig. D1 but for O₃.

**Table E1: CV results of the reanalysis (outside bracket) and base simulation (in bracket) for PM$_{2.5}$ concentrations in**
**different regions of China at different temporal scales**

| PM$_{2.5}$ ($\mu g/m^3$) | NCP | | | | NE | | | |
|---|---|---|---|---|---|---|---|---|
| | $R^2$ | MBE | NMB(%) | RMSE | $R^2$ | MBE | NMB (%) | RMSE |
| Hourly | 0.85 (0.33) | -3.3 (22.4) | -4.8 (32.8) | 25.1 (62.6) | 0.77 (0.25) | -2.6 (2.8) | -5.8 (6.5) | 22.6 (44.5) |
| Daily | 0.90 (0.44) | -3.4 (22.3) | -4.9 (32.4) | 17.5 (51.2) | 0.86 (0.32) | -2.6 (2.6) | -5.9 (6.0) | 14.7 (35.1) |
| Monthly | 0.92 (0.56) | -3.4 (22.2) | -4.9 (32.4) | 11.4 (34.1) | 0.86 (0.38) | -2.6 (2.7) | -5.9 (6.0) | 9.7 (21.4) |
| Yearly | 0.92 (0.56) | -3.6 (20.8) | -5.0 (29.2) | 8.7 (27.3) | 0.79 (0.35) | -3.1 (0.4) | -6.6 (0.8) | 8.8 (16.7) |
| | SE | | | | SW | | | |
| | $R^2$ | MBE | NMB(%) | RMSE | $R^2$ | MBE | NMB (%) | RMSE |
| Hourly | 0.85 (0.25) | -1.8 (22.2) | -3.8 (47.6) | 14.9 (51.5) | 0.79 (0.22) | -1.4 (30.3) | -3.4 (74.7) | 16.5 (57.4) |
| Daily | 0.90 (0.31) | -1.8 (22.2) | -3.8 (47.4) | 10.6 (45.4) | 0.86 (0.29) | -1.4 (30.0) | -3.4 (74.2) | 12.1 (51.6) |
| Monthly | 0.92 (0.45) | -1.8 (22.1) | -3.8 (47.2) | 7.4 (33.7) | 0.86 (0.49) | -1.5 (29.8) | -3.7 (73.3) | 9.7 (42.8) |
| Yearly | 0.90 (0.37) | -2.0 (20.5) | -4.0 (42.0) | 6.1 (29.3) | 0.79 (0.47) | -2.2 (27.2) | -5.0 (63.2) | 9.5 (38.8) |
| | NW | | | | Central | | | |
| | $R^2$ | MBE | NMB(%) | RMSE | $R^2$ | MBE | NMB (%) | RMSE |
| Hourly | 0.52 (0.11) | -7.3 (-28.7) | -13.1 (-51.1) | 52.1 (73.0) | 0.72 (0.23) | -4.1 (0.8) | -8.2 (1.6) | 26.6 (47.5) |
| Daily | 0.66 (0.15) | -7.5 (-29.0) | -13.2 (-51.3) | 39.4 (66.0) | 0.83 (0.30) | -4.2 (0.7) | -8.3 (1.4) | 19.1 (39.9) |
| Monthly | 0.72 (0.28) | -7.4 (-28.9) | -13.1 (-51.3) | 26.9 (50.3) | 0.85 (0.42) | -4.2 (0.7) | -8.2 (1.4) | 13.1 (26.1) |
| Yearly | 0.64 (0.40) | -9.8 (-33.5) | -16.1 (-54.9) | 23.5 (43.1) | 0.77 (0.31) | -5.4 (-3.6) | -10.1 (-6.7) | 12.5 (24.3) |

**Table E2: CV results of the reanalysis (outside bracket) and base simulation (in bracket) for PM$_{10}$ concentrations in**

**different regions of China at different temporal scales**

| PM$_{10}$ | NCP | | | | NE | | | |
|---|---|---|---|---|---|---|---|---|
| ($\mu$g/m$^3$) | R$^2$ | MBE | NMB(%) | RMSE | R$^2$ | MBE | NMB (%) | RMSE |
| Hourly | 0.79 (0.23) | -7.7 (-14.6) | -6.4 (-12.1) | 43.7 (88.3) | 0.71 (0.18) | -7.6 (-23.6) | -9.6 (-29.8) | 39.8 (70.8) |
| Daily | 0.86 (0.31) | -7.6 (-14.2) | -6.3 (-11.7) | 30.9 (71.8) | 0.79 (0.25) | -7.6 (-23.6) | -9.7 (-30.0) | 27.1 (56.8) |
| Monthly | 0.86 (0.38) | -7.6 (-14.2) | -6.3 (-11.8) | 21.4 (44.9) | 0.76 (0.29) | -7.7 (-23.6) | -9.8 (-30.0) | 19.4 (39.6) |
| Yearly | 0.85 (0.46) | -7.6 (-15.8) | -6.2 (-12.8) | 17.6 (33.0) | 0.67 (0.31) | -8.3 (-26.5) | -10.3 (-32.6) | 18.4 (36.2) |

| | SE | | | | SW | | | |
|---|---|---|---|---|---|---|---|---|
| | R$^2$ | MBE | NMB(%) | RMSE | R$^2$ | MBE | NMB (%) | RMSE |
| Hourly | 0.77 (0.18) | -4.4 (6.9) | -5.9 (9.4) | 26.0 (61.2) | 0.69 (0.15) | -5.1 (13.0) | -7.5 (19.1) | 30.2 (66.2) |
| Daily | 0.85 (0.23) | -4.1 (8.1) | -5.6 (11.1) | 18.6 (52.0) | 0.77 (0.21) | -5.0 (13.1) | -7.4 (19.6) | 22.4 (56.5) |
| Monthly | 0.85 (0.38) | -4.2 (7.5) | -5.7 (10.2) | 13.7 (33.3) | 0.76 (0.38) | -5.2 (12.5) | -7.8 (18.5) | 18.7 (41.4) |
| Yearly | 0.81 (0.36) | -4.7 (4.9) | -6.1 (6.5) | 12.3 (26.3) | 0.62 (0.38) | -6.8 (8.7) | -9.6 (12.2) | 19.3 (35.7) |

| | NW | | | | Central | | | |
|---|---|---|---|---|---|---|---|---|
| | R$^2$ | MBE | NMB(%) | RMSE | R$^2$ | MBE | NMB (%) | RMSE |
| Hourly | 0.46 (0.08) | -21.5 (-88.5) | -18.0 (-74.1) | 105.5 (150.2) | 0.61 (0.11) | -14.6 (-45.6) | -14.1 (-43.9) | 57.3 (96.4) |
| Daily | 0.56 (0.11) | -21.5 (-89.3) | -17.9 (-74.1) | 85.5 (141.6) | 0.72 (0.14) | -14.6 (-45.5) | -14.1 (-43.8) | 42.1 (84.6) |
| Monthly | 0.59 (0.17) | -20.8 (-89.5) | -17.2 (-74.0) | 64.0 (118.9) | 0.74 (0.28) | -14.6 (-45.3) | -14.1 (-43.8) | 30.2 (62.5) |
| Yearly | 0.58 (0.23) | -23.8 (-92.3) | -19.3 (-74.7) | 55.8 (110.2) | 0.67 (0.25) | -16.4 (-50.1) | -15.4 (-46.8) | 28.0 (60.4) |

**Table E3: CV results of the reanalysis (outside bracket) and base simulation (in bracket) for SO₂ concentrations in**

**different regions of China at different temporal scales**

| SO₂ (μg/m³) | NCP | | | | NE | | | |
|---|---|---|---|---|---|---|---|---|
| | $R^2$ | MBE | NMB(%) | RMSE | $R^2$ | MBE | NMB (%) | RMSE |
| Hourly | 0.62 (0.10) | -3.6 (26.4) | -9.4 (69.4) | 31.5 (63.1) | 0.46 (0.08) | -2.0 (5.1) | -6.9 (17.5) | 34.8 (53.2) |
| Daily | 0.74 (0.16) | -3.6 (26.4) | -9.4 (69.6) | 22.8 (52.7) | 0.62 (0.13) | -2.0 (5.1) | -7.0 (17.6) | 23.8 (42.2) |
| Monthly | 0.79 (0.19) | -3.7 (26.2) | -9.6 (68.4) | 17.1 (43.6) | 0.71 (0.14) | -2.0 (5.0) | -6.9 (17.3) | 17.9 (34.9) |
| Yearly | 0.81 (0.18) | -4.2 (23.7) | -10.2 (56.9) | 13.3 (36.1) | 0.56 (0.14) | -2.4 (2.7) | -7.6 (8.7) | 15.9 (27.7) |

| | SE | | | | SW | | | |
|---|---|---|---|---|---|---|---|---|
| | $R^2$ | MBE | NMB(%) | RMSE | $R^2$ | MBE | NMB (%) | RMSE |
| Hourly | 0.42 (0.01) | -1.0 (29.8) | -5.7 (169.6) | 14.6 (69.3) | 0.27 (0.01) | -1.9 (44.2) | -12.1 (277.2) | 16.7 (88.1) |
| Daily | 0.55 (0.01) | -1.0 (29.9) | -5.7 (170.2) | 10.5 (63.3) | 0.38 (0.01) | -1.9 (44.1) | -12.2 (276.5) | 11.8 (80.3) |
| Monthly | 0.61 (0.01) | -1.0 (29.7) | -5.7 (168.6) | 7.8 (55.8) | 0.46 (0.02) | -2.0 (43.9) | -12.4 (273.7) | 9.1 (73.7) |
| Yearly | 0.66 (0.01) | -1.4 (28.0) | -7.1 (144.5) | 7.9 (52.7) | 0.53 (0.01) | -2.7 (41.2) | -15.2 (231.3) | 9.5 (68.3) |

| | NW | | | | Central | | | |
|---|---|---|---|---|---|---|---|---|
| | $R^2$ | MBE | NMB(%) | RMSE | $R^2$ | MBE | NMB (%) | RMSE |
| Hourly | 0.31 (0.01) | -0.3 (9.4) | -2.3 (61.6) | 22.7 (40.4) | 0.30 (0.02) | -4.4 (13.2) | -17.0 (51.3) | 36.0 (58.9) |
| Daily | 0.42 (0.01) | -0.3 (9.4) | -1.8 (62.2) | 17.8 (36.2) | 0.49 (0.03) | -4.4 (13.2) | -17.0 (51.5) | 23.6 (49.1) |
| Monthly | 0.48 (0.03) | -0.3 (9.3) | -2.2 (61.1) | 13.4 (30.3) | 0.59 (0.03) | -4.4 (13.1) | -17.0 (51.0) | 18.2 (43.2) |
| Yearly | 0.29 (0.00) | -1.9 (6.6) | -10.5 (35.9) | 15.8 (28.0) | 0.50 (0.00) | -5.6 (8.6) | -19.0 (29.3) | 18.6 (40.2) |

**Table E4: CV results of the reanalysis (outside bracket) and base simulation (in bracket) for NO₂ concentrations in**

**different regions of China at different temporal scales**

| $NO_2$ ($\mu g/m^3$) | NCP | | | | NE | | | |
|---|---|---|---|---|---|---|---|---|
| | $R^2$ | MBE | NMB(%) | RMSE | $R^2$ | MBE | NMB (%) | RMSE |
| Hourly | 0.67 (0.20) | -1.4 (-3.0) | -3.5 (-7.1) | 16.8 (26.5) | 0.61 (0.27) | -1.6 (-5.9) | -5.0 (-19.1) | 15.8 (22.4) |
| Daily | 0.72 (0.22) | -1.4 (-2.9) | -3.3 (-7.1) | 12.4 (20.8) | 0.66 (0.34) | -1.5 (-5.9) | -4.9 (-19.0) | 11.7 (17.2) |
| Monthly | 0.72 (0.24) | -1.4 (-2.9) | -3.3 (-7.1) | 9.3 (15.5) | 0.64 (0.37) | -1.5 (-5.9) | -5.0 (-19.1) | 9.3 (13.7) |
| Yearly | 0.67 (0.36) | -1.4 (-3.8) | -3.3 (-9.0) | 7.5 (11.0) | 0.64 (0.45) | -1.5 (-6.4) | -4.8 (-20.3) | 7.8 (11.7) |

| | SE | | | | SW | | | |
|---|---|---|---|---|---|---|---|---|
| | $R^2$ | MBE | NMB(%) | RMSE | $R^2$ | MBE | NMB (%) | RMSE |
| Hourly | 0.64 (0.23) | -1.9 (-1.3) | -5.9 (-4.0) | 14.9 (24.1) | 0.49 (0.19) | -3.9 ( -9.9) | -14.0 (-35.7) | 16.4 (23.2) |
| Daily | 0.71 (0.28) | -1.8 (-1.3) | -5.8 (-4.0) | 11.2 (19.2) | 0.55 (0.28) | -3.9 ( -9.9) | -14.0 (-35.7) | 12.8 (18.7) |
| Monthly | 0.72 (0.36) | -1.8 (-1.2) | -5.8 (-3.9) | 8.8 (14.7) | 0.48 (0.32) | -4.0 (-10.0) | -14.4 (-36.0) | 12.6 (17.2) |
| Yearly | 0.66 (0.49) | -1.9 (-2.2) | -6.0 (-6.6) | 7.8 (11.7) | 0.46 (0.37) | -4.6 (-11.0) | -16.1 (-38.7) | 11.8 (16.4) |

| | NW | | | | Central | | | |
|---|---|---|---|---|---|---|---|---|
| | $R^2$ | MBE | NMB(%) | RMSE | $R^2$ | MBE | NMB (%) | RMSE |
| Hourly | 0.46 (0.20) | -4.3 (-18.0) | -12.9 (-54.4) | 24.3 (31.5) | 0.50 (0.20) | -5.2 (-16.7) | -15.1 (-48.3) | 20.5 (29.2) |
| Daily | 0.55 (0.27) | -4.1 (-18.0) | -12.5 (-54.4) | 18.3 (27.0) | 0.58 (0.28) | -5.2 (-16.7) | -15.0 (-48.3) | 15.4 (24.7) |
| Monthly | 0.59 (0.40) | -4.2 (-18.0) | -12.7 (-54.3) | 15.3 (23.8) | 0.61 (0.40) | -5.2 (-16.6) | -15.1 (-48.3) | 12.8 (21.6) |
| Yearly | 0.40 (0.36) | -6.0 (-19.7) | -17.3 (-56.3) | 16.5 (23.8) | 0.55 (0.36) | -5.6 (-17.6) | -16.0 (-50.1) | 12.2 (21.4) |

**Table E5: CV results of the reanalysis (outside bracket) and base simulation (in bracket) for CO concentrations in**

**different regions of China at different temporal scales**

| CO (mg/m$^3$) | NCP | | | | NE | | | |
|---|---|---|---|---|---|---|---|---|
| | $R^2$ | MBE | NMB(%) | RMSE | $R^2$ | MBE | NMB (%) | RMSE |
| Hourly | 0.67 (0.25) | -0.03 (-0.59) | -2.49 (-43.4) | 0.64 (1.13) | 0.50 (0.20) | -0.05 (-0.51) | -5.3 (-51.9) | 0.59 (0.88) |
| Daily | 0.72 (0.31) | -0.03 (-0.59) | -2.15 (-43.3) | 0.50 (0.99) | 0.56 (0.25) | -0.05 (-0.51) | -4.9 (-51.7) | 0.46 (0.78) |
| Monthly | 0.74 (0.34) | -0.03 (-0.59) | -2.24 (-43.5) | 0.38 (0.85) | 0.59 (0.25) | -0.05 (-0.51) | -5.2 (-52.0) | 0.37 (0.70) |
| Yearly | 0.71 (0.14) | -0.04 (-0.64) | -2.75 (-45.1) | 0.32 (0.85) | 0.55 (0.14) | -0.06 (-0.56) | -5.9 (-54.0) | 0.35 (0.74) |

| | SE | | | | SW | | | |
|---|---|---|---|---|---|---|---|---|
| | $R^2$ | MBE | NMB(%) | RMSE | $R^2$ | MBE | NMB (%) | RMSE |
| Hourly | 0.42 (0.13) | -0.06 (-0.36) | -6.4 (-38.2) | 0.39 (0.62) | 0.36 (0.07) | -0.08 (-0.32) | -9.4 (-36.4) | 0.46 (0.65) |
| Daily | 0.45 (0.15) | -0.06 (-0.36) | -6.1 (-38.0) | 0.34 (0.57) | 0.40 (0.08) | -0.08 (-0.31) | -9.1 (-36.3) | 0.39 (0.59) |
| Monthly | 0.44 (0.14) | -0.06 (-0.36) | -6.2 (-38.1) | 0.28 (0.51) | 0.40 (0.08) | -0.08 (-0.32) | -9.4 (-36.7) | 0.34 (0.54) |
| Yearly | 0.38 (0.05) | -0.06 (-0.38) | -6.5 (-39.3) | 0.25 (0.50) | 0.36 (0.01) | -0.09 (-0.36) | -10.1 (-39.1) | 0.36 (0.57) |

| | NW | | | | Central | | | |
|---|---|---|---|---|---|---|---|---|
| | $R^2$ | MBE | NMB(%) | RMSE | $R^2$ | MBE | NMB (%) | RMSE |
| Hourly | 0.38 (0.12) | -0.19 (-1.02) | -15.0 (-79.3) | 1.13 (1.55) | 0.44 (0.22) | -0.13 (-0.76) | -11.2 (-65.2) | 0.73 (1.11) |
| Daily | 0.45 (0.18) | -0.19 (-1.01) | -14.6 (-79.2) | 0.92 (1.43) | 0.49 (0.27) | -0.13 (-0.76) | -10.8 (-65.1) | 0.62 (1.04) |
| Monthly | 0.50 (0.29) | -0.19 (-1.02) | -15.1 (-79.3) | 0.75 (1.32) | 0.53 (0.32) | -0.13 (-0.76) | -11.1 (-65.2) | 0.52 (0.97) |
| Yearly | 0.13 (0.12) | -0.31 (-1.18) | -21.1 (-80.8) | 0.85 (1.35) | 0.19 (0.08) | -0.17 (-0.84) | -13.3 (-67.3) | 0.69 (1.08) |

**Table E6: CV results of the reanalysis (outside bracket) and base simulation (in bracket) for O₃ concentrations in**
**different regions of China at different temporal scales**

| O₃ (μg/m³) | NCP | | | | NE | | | |
|---|---|---|---|---|---|---|---|---|
| | $R^2$ | MBE | NMB(%) | RMSE | $R^2$ | MBE | NMB (%) | RMSE |
| Hourly | 0.83 (0.50) | -3.9 (-24.5) | -6.1 (-39.1) | 22.1 (44.3) | 0.76 (0.38) | -3.6 (-15.4) | -6.0 (-25.6) | 21.0 (36.7) |
| Daily | 0.83 (0.48) | -3.8 (-24.5) | -6.0 (-39.1) | 16.3 (37.0) | 0.76 (0.34) | -3.6 (-15.3) | -5.9 (-25.5) | 16.4 (30.8) |
| Monthly | 0.85 (0.62) | -3.8 (-24.4) | -6.1 (-39.1) | 12.6 (31.6) | 0.76 (0.42) | -3.6 (-15.2) | -5.9 (-25.4) | 13.1 (25.2) |
| Yearly | 0.72 (0.29) | -3.7 (-23.1) | -6.2 (-38.6) | 9.2 (26.8) | 0.62 (0.18) | -3.5 (-14.4) | -6.1 (-25.0) | 10.0 (20.4) |

| | SE | | | | SW | | | |
|---|---|---|---|---|---|---|---|---|
| | $R^2$ | MBE | NMB(%) | RMSE | $R^2$ | MBE | NMB (%) | RMSE |
| Hourly | 0.77 (0.57) | -2.3 (-10.0) | -3.9 (-17.4) | 21.1 (38.2) | 0.69 (0.40) | 0.8 (9.6) | 1.4 (18.0) | 22.2 (33.6) |
| Daily | 0.69 (0.44) | -2.2 (-10.0) | -3.8 (-17.3) | 15.8 (31.0) | 0.64 (0.34) | 0.8 (9.7) | 1.5 (18.1) | 17.4 (26.7) |
| Monthly | 0.69 (0.43) | -2.2 (-10.0) | -3.9 (-17.3) | 12.4 (24.2) | 0.64 (0.44) | 0.8 (9.7) | 1.6 (18.1) | 14.4 (21.1) |
| Yearly | 0.45 (0.07) | -2.4 (-10.1) | -4.2 (-17.7) | 10.0 (20.7) | 0.42 (0.28) | 1.3 (10.0) | 2.6 (19.4) | 12.0 (17.7) |

| | NW | | | | Central | | | |
|---|---|---|---|---|---|---|---|---|
| | $R^2$ | MBE | NMB(%) | RMSE | $R^2$ | MBE | NMB (%) | RMSE |
| Hourly | 0.52 (0.31) | -2.7 (-2.2) | -4.6 (-3.8) | 28.3 (33.2) | 0.71 (0.45) | -1.5 (-0.8) | -2.5 (-1.3) | 23.9 (32.5) |
| Daily | 0.50 (0.31) | -2.6 (-2.1) | -4.5 (-3.6) | 22.9 (26.6) | 0.67 (0.42) | -1.4 (-0.7) | -2.4 (-1.1) | 17.8 (23.9) |
| Monthly | 0.58 (0.42) | -2.6 (-2.1) | -4.5 (-3.6) | 19.1 (22.1) | 0.72 (0.56) | -1.4 (-0.7) | -2.4 (-1.2) | 13.9 (17.6) |
| Yearly | 0.37 (0.24) | -1.6 (-0.6) | -2.9 (-1.1) | 15.7 (17.1) | 0.53 (0.30) | -0.8 (0.2) | -1.4 (0.4) | 11.6 (14.1) |

**Author contributions**

X.T., J.Z., and Z.W. conceived and designed the project; H.W., L.K., X.T., and L.W. established the data assimilation system; Q.W. and L.K. performed the meteorology simulations; X.T., L.K., H.C., H.W., H.Z., G.J. and M.L. conducted the ensemble simulations with the NAQPMS model; J.L., L.Z., W.W., B.L., Q.W., D.C. and T.S. provided the air quality monitoring data; W.H. executed the quality control of the observation data; F.L. estimated the representativeness error of the observations; and L.K. carried out the CAQRA calculations, generated the figures and wrote the paper with comments provided by G.C.

**Competing interests**

The authors declare that they have no conflicts of interest.

**Acknowledgements**

This study was funded by the National Natural Science Foundation (Grant Nos. 91644216, 41575128, and 41875164), the CAS Strategic Priority Research Program (Grant No. XDA19040201), and the CAS Information Technology Program (Grant No. XXH13506-302). This study was supported by the National Key Scientific and Technological Infrastructure project "Earth System Science Numerical Simulator Facility" (EarthLab). We would like to thank the editor and the two reviewers for their valuable comments.

[revised manuscript text omitted]